# Reconciling chemical flame retardant exposure and fire risk in domestic furniture

**Paul Whaley**[1,2]*, **Stephen Wattam**[3], **Clare Bedford**[4], **Nia Bell**[5], **Stuart Harrad**[6], **Nicola Jones**[4], **Thomas Kirkbride**[7], **Dzhordzhio Naldzhiev**[7], **Elena Payne**[5], **Elli-Jo Wooding**[4], **T. Richard Hull**[4]*

**1** Lancaster Environment Centre, Lancaster University, Lancaster, United Kingdom, **2** WhaleyResearch, Leuchars, Fife, United Kingdom, **3** W&P Academic Consultancy Limited, Northallerton, United Kingdom, **4** Centre for Fire and Hazards Science, University of Central Lancashire, Preston, United Kingdom, **5** Oakdene Hollins, Aylesbury, United Kingdom, **6** School of Geography, Earth, and Environmental Sciences, University of Birmingham, Birmingham, United Kingdom, **7** Office for Product Safety and Standards, London, United Kingdom

* p.whaley@lancaster.ac.uk (PW); trhull@uclan.ac.uk (TRH)

## Abstract

### Introduction

Evidence suggests that standards for resistance of furniture to ignition may lead to an increase in use of chemical flame retardants (CFRs). This is motivating the development of new approaches that maintain high levels of fire safety while facilitating a reduction in use of CFRs. However, reconciling potential fire risk with use of CFRs in relation to specific policy objectives is challenging.

### Objectives

To inform the development of a new policy in the UK for the fire safety of furniture, we developed for domestic furniture quantitative models of fire risk and potential for CFR exposure. We then combined the models to determine if any lower fire risk, higher CFR exposure categories of furniture were identifiable.

### Methods

We applied a novel mixed-methods approach to modelling furniture fire risk and CFR exposure in a data-poor environment, using literature-based concept mapping, qualitative research, and data visualisation methods to generate fire risk and CFR exposure models and derive furniture product rankings.

### Results

Our analysis suggests there exists a cluster of furniture types including baby and infant products and pillows that have comparable overall properties in terms of lower fire risk and higher potential for CFR exposure.

**Data Availability Statement:** All supplemental materials can be accessed at https://doi.org/10.5281/zenodo.8351890.

**Funding:** The research was funded under competitive tender "CR21021 - Fire Risks of

Upholstered Products" as part of the UK Department for Department for Business, Energy & Industrial Strategy (BEIS) Office for Product Safety and Standards (OPSS) Strategic Research Programme. Two BEIS employees from the Research, Evidence and Dissemination team within OPSS (TK, DN) were involved in study design and preparation of the manuscript. The manuscript went through an internal clearance process at BEIS and OPSS before being approved for submission.

**Competing interests:** PW declares they are a self-employed consultant who provides research, training, and editorial services on a contracted basis for industry, NGO, academic, and publishing clients. Given the controversial nature of the issues researched in this project, the way the work is received could affect their prospects for securing future contracts in this area. However, they do not identify any specific financial interests that would be directly affected at this time. For non-financial interests, they declare involvement in environmental advocacy for over 15 years and being motivated by a perceived need for significant improvement in identification and management of potential risks to health posed by the environment in which humans live. They have written several papers and held public positions, such as developing and/or signing consensus statements, about the need for improved regulation and scientific assessment of chemical substances with flame retardants as an example. They are actively involved with NGOs, particularly in the UK, who are advocating for reforms to chemical regulation. Being prominently involved as a Research Fellow in the Evidence-Based Toxicology Collaboration, their views could be perceived as upholding that organisation's views. They are also Editor-in-Chief of Evidence-Based Toxicology, a new journal concerned with promoting improved research and publishing practices in toxicology and environmental health. While these financial and non-financial interests are pertinent to the manuscript, PW does not believe them to have compromised the integrity of the work undertaken. SW declares scientific consultancy services in fields unrelated to the topic of research in this manuscript. The way the work is received could affect their prospects of future employment. CB declares employment at the University of Central Lancashire in the Centre for Fire and Hazards Sciences as part of a team who examine the toxicity of flammable materials. CB is also a trade union representative at the University. As a result of previous work in their current role, CB declares an awareness of the potential hazards presented by both flammable materials and flame retardants, and the belief that it is important to accurately

## Discussion

There are multiple obstacles to reconciling fire risk and CFR use in furniture. In particular, these include a lack of empirical data that would allow absolute fire risk and exposure levels to be quantified. Nonetheless, it seems that our modelling method can potentially yield meaningful product clusters, providing a basis for further research.

## Introduction

From 1960 to 1980 UK fire deaths doubled from 506 to 1035, alongside the replacement of furniture containing natural fillings with flexible polyurethane foam, with most of the increase being attributed to inhalation of toxic smoke [1]. In response, the UK enacted the Furniture and Furnishings (Fire) (Safety) Regulations (1988) (FFRs). To date, these are some of the world's strictest regulations controlling flammability of upholstered furniture. They require both the fabric and filling used to upholster furniture to resist ignition from small sources such as a match, cigarette, or the equivalent of four sheets of burning newspaper [2]. For most manufacturers, these ignition requirements are met by incorporation of chemical flame retardants (CFRs) into furniture upholstery, as a coating behind the fabric [3], and within the filling [4]. In comparison, regulations in the European Union and the United States specify limits to flammability in a way that does not drive the addition of flame retardants to furniture fillings [5].

As CFRs are significantly more expensive than the polymer they are used to protect, they are unlikely to be present at loadings greater than those needed to pass the regulatory test. While the use of CFRs results in a reduction in the ignitability of upholstery, it has also led to UK household dust having some of the world's highest concentrations of CFRs [6, 7], many of which are known to be harmful [8–12]. Once upholstery has been ignited, CFRs have minimal effect on the burning rate of the fabric, filling or furniture item.

A wide range of compounds are sold as flame retardants. These can be classified by their mode of action, or composition. Most flame retardants act in one of the following ways:

1. In the gas phase as radical flame quenchers. These usually contain chlorine, bromine or organophosphorus compounds. By stopping the flame reactions midway, they may increase the yield of products of incomplete combustion, including smoke and the asphyxiants carbon monoxide and hydrogen cyanide [13].

2. As mineral fillers, releasing inert gases such as water or carbon dioxide by endothermic decomposition. The most common are aluminium hydroxide and magnesium hydroxide. Typically, their flame retardant effect is only significant at loadings above 50% of the host polymer [14]. There is evidence that mineral fillers have less adverse effect on smoke toxicity [15].

3. As char formers and intumescents, forming a non-combustible protective layer on the surface of the burning material, so fire growth can be slowed or stopped. Intumescents produce a swollen, protective layer above the surface. Most char-forming and intumescent flame retardants involve multicomponent formations. By reducing the flow of volatiles to the gas phase, they will also reduce the amount of smoke and hence its toxic effects.

Upholstered furniture consists of open weave fabrics and open, porous fillings, which are the most difficult to protect from fire. Their low thermal inertia (the product of the heat capacity, thermal conductivity and density, (krC)) means that they are easily ignited and burn

assess hazards and strive to protect people from them as much as possible. They do not think that this conviction has affected their ability to contribute objectively to this work. NB declares employment by Oakdene Hollins at the time the research was carried out. Oakdene Hollins is a sustainability consultancy that works with private and public sector clients on reducing their environmental impacts. SH declares being an academic researcher. Given the controversial nature of the issues we have researched in this project, the way the work is received could affect their prospects for securing future research grants and contracts, as well external consultancy work in this area. However, they cannot identify any specific financial interests that would be directly affected at this time. For non-financial interests, their extensive previous research into human exposure to a wide range of chemical contaminants including flame retardants inherently influences the submitted publication. They declare being a current member of the UK government's Hazardous Substances Advisory Committee; while this manuscript may be perceived as reflecting the views of this committee, it has not to date involved consideration of human exposure to flame retardants and moreover SH's views are purely their own and not those of the committee. While these financial and non-financial interests are pertinent to the manuscript, SH does not believe them to have compromised the integrity of the work undertaken. NJ declares employment as a Post Doctoral Researcher at the University of Central Lancashire at the time the research was carried out. TK declares they are currently employed by the Department for Business and Trade (formerly the Department for Business, Energy, and Industrial Strategy at the time this research was completed), and that they commissioned this work on behalf of the Office for Product Safety and Standards under the science division (Research, Evidence and Dissemination team) within OPSS. They were involved in setting the objectives and the design of the research. TK's role in this context was to provide scientific and technical advice to policy makers, and they were not themselves a decision-maker or policy lead. As a civil servant they state they are impartial to the findings of the research, and that the research does not necessarily represent the views of the department. DN declares being employed by OPSS during the conduct of the study, as Head of the Science Strategy team overseeing research projects in the science division (Research, Evidence and Dissemination team) within OPSS. DN's role in this context was to provide scientific and technical advice to policy makers, and they

quickly. They usually rely on gas flame quenchers for fire protection. Currently, the most widely used CFRs in domestic furniture include decabromodiphenyl ethane (DBDPE) (predominantly mixed at around 50% loading into a latex backcoating applied to the underside of upholstery fabric), and tris (2-chloropropyl) phosphate (TCIPP) (incorporated into flexible polyurethane foam). DBDPE has a similar structure to the flame retardants decabromodiphenylether (Deca-BDE) and polybromobiphenyl, both of which were found to be persistent, bioaccumulative, and toxic (PBT) and withdrawn from use [16]. The structural similarity of DBDPE to Deca-BDE suggests that it too will be withdrawn at some point in the near future: it is currently under assessment as PBT [17]. TCIPP is currently on the candidate list of substances of very high concern (SVHC) under the European REACH regulations.

UK regulations for the fire safety of upholstered furniture are currently under review by the UK Government. As part of this review process, we were asked to propose a methodological framework to assess risks and benefits of including certain peripheral product types, such as garden furniture, baby products, cushions, headboards etc., into the revised UK regulations, as it relates to potential for fire risk and exposure to CFRs. The current review of the FFRs is timely. Furniture construction, smoking habits, the presence of smoke alarms and the use of open flame heating have all changed radically in the 40 years since the tests and regulations were developed. Meanwhile, a general need to reduce the presence of toxic substances in our environment, the need to recycle products such as furniture at their end-of-life, and the unsuitability of furniture containing CFRs for landfill or material reclamation, are much higher priorities than they were in the 1980s.

## Objectives and scope

The overall aim of the work was to produce a conceptual framework against which different furniture product types could be assessed to inform proposals relating to the scope of the new regulations. The framework should allow for upholstered domestic furniture the potential for exposure to CFRs to be balanced against risks from fire.

This work is premised on the desirability of reducing exposure to CFRs. One way to prevent exposure is to completely eliminate them from use. However, many materials used in furniture are inherently flammable. Until this is addressed, complete elimination of FRs may pose unacceptable fire risks. This project aims to identify clusters of furniture types where the balance of fire risk is low and potential for CFR exposure high, such that these types could potentially be exempted from UK FFRs (such as by removing the requirement to pass ignition tests).

The specific objectives of this research were to:

1. Develop a quantitative model of risk of personal injury or property damage from a fire that originates in an item of domestic furniture (henceforth, "Fire Risk Model"), and rank a selection of furniture types on this model;

2. Develop a quantitative model of potential CFR exposure from domestic use of an item of furniture (henceforth, "Exposure Model"), and rank the selection of furniture types on this model;

3. Reconcile the fire risk and CFR exposure model rankings to determine if any coherent clusters of furniture types can be identified that may be informative to the development and policy goals of the FFRs.

In this paper, we present the methods used to (a) develop the Fire Risk and CFR Exposure Models, and (b) reconcile the two models to identify clusters of furniture types. We describe constraints in the available data, our simplifying assumptions, the implications of both for

were not themselves a decision-maker or policy lead. EP declares employment by Oakdene Hollins at the time the research was carried out. Oakdene Hollins is a sustainability consultancy that works with private and public sector clients on reducing their environmental impacts. EW declares employment by the University of Central Lancashire at the time the research was carried out. RH declares employment as a Professor of Chemistry and Fire Science by the University of Central Lancashire since 2007. They do not have any secondary employment, consultancy, board membership, patents or patent applications. They have received research funding from external organisations: in 2018, £15,000 by Silentnight Beds Ltd., to undertake four large scale fire tests and assess the smoke toxicity; in 2019, £161,000 from Innovate UK and Silentnight Beds Ltd., to support a Knowledge Transfer Partnership aimed at helping Silentnight reduce their fire retardant use, the smoke toxicity, and the recyclability of their products, in order to improve their access to the wider mainland European market (Silentnight Beds had no involvement in the current BEIS project, nor in the authorship of this paper); funding from Fire Safe Europe to support a PhD student investigating smoke toxicity of construction products; and funding from the Construction Products Group, Europe to develop fire protective coatings for structural steel. RH has also supported their colleague Prof Anna Stec in their role as an expert witness to the United Kingdom's Grenfell Tower Inquiry, and is a member of the BEIS Expert Advisory Panel for the revision of the English Furniture Flammability Regulations. RH does not believe they have any interests that compete with those of the research described in the manuscript.

appropriately interpreting the model outputs, and describe how the models could be improved. Our approach involves a number of methodological approaches not previously applied in the fire sciences. These have potential implications for the assessment and management of furniture fire risks that should be of general interest to the scientific community. We hope the detailed description of our methods along with comprehensive supplemental materials will be of value to anyone seeking to validate or extend our approach.

Within this work, "fire risk" is defined as the risk of a fire originating in an item of domestic furniture that is severe enough to cause injury to someone currently using that item of furniture. This paper presents for a scientific audience some of the research described in the UK Office for Product Safety and Standards (OPSS) report *Fire risks of upholstered products*. The OPSS report includes further information about CFR technologies use in upholstered furniture and provides statistical analysis of international incidence of fires and fire deaths [18]. Data and supplemental information (SM) for this study are available at DOI 10.5281/zenodo.8351890.

## Operational constraints

We were specifically tasked with modelling CFR exposure from domestic furniture while the furniture is in use in a home environment, and with modelling risk of injury to person or damage to property from a fire that originates in an item of domestic furniture. The modelling was to be developed for furniture categories (product types) rather than individual items. Factors such as smoke opacity, smoke toxicity and problems stemming from the presence or release of CFRs at end-of-life were outside the scope of the current work. We do not assess the effectiveness of CFRs or any other intervention for mitigating fire risk.

To provide evidence in a timely manner for the relevant policy processes, it was not possible to systematically review the literature around CFR exposure and the role of furniture in domestic fires (over 10,000 relevant documents) or to conduct experiments to develop new empirical data. Therefore, we selected model development and data collection methods that worked with manageable literature samples, accommodating or acknowledging data gaps as appropriate, while being sufficiently grounded in data and expert opinion that we still present meaningful product rankings as per the objectives of the project.

## Methods

### Summary of the methodology

Our methodology consisted of three broad steps: developing the fire risk and CFR exposure models; populating the models with data; and developing the rank orderings for products within each model and providing an overall reconciled product ranking (Fig 1).

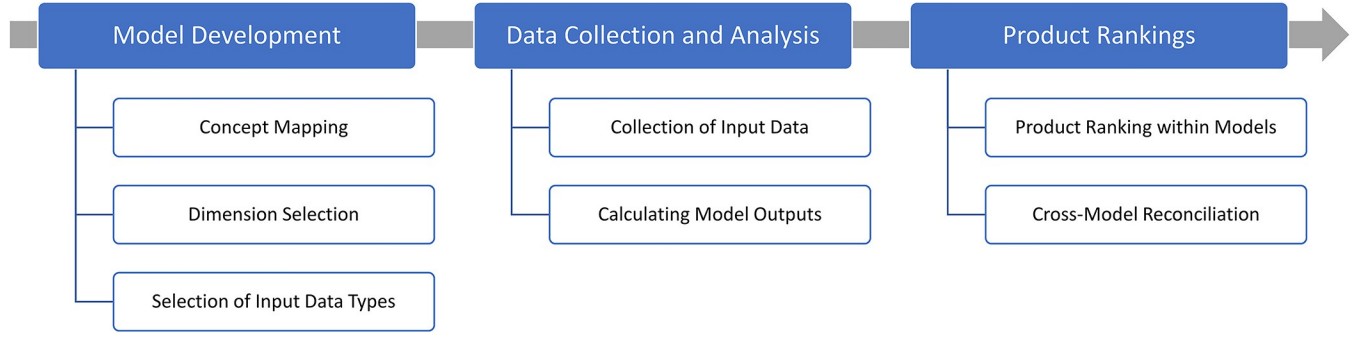

**Fig 1. Summary of the methodology for developing the product rankings.**

## Model development

Our model development process consisted of two steps: providing a comprehensive map of factors that are considered by domain experts to potentially affect fire risk and potential CFR exposure in domestic furnishings (a "concept network"); then, selecting from the concept network factors that can function as dimensions for fire risk and exposure models. This is an approach based on grounded theory [19], modified for the timeline and budget of the project.

**Concept mapping.** We needed to identify factors that affect fire risk and potential CFR exposure in domestic furnishings in a way that is recognisably grounded in current expert understanding of the issues, without conducting an exhaustive review of the literature or interviewing large numbers of fire safety experts. We therefore decided to analyse literature reviews about furniture and flame retardants. This decision was based on two assumptions: (1) that literature reviews are where scientists present in detail their expert view as to how furniture poses fire risks and potential for CFR exposure; and (2) since literature reviews are summaries of primary studies, the concepts being discussed in the reviews will be a relatively complete representation of the concepts being discussed in the primary literature. Since our objective at this stage was only to develop a comprehensive list of factors that could be included in a model, not weight them for contribution to model results, we believe this method was appropriate.

We note that narrative descriptions of fires in reports by fire investigators may provide useful data for concept mapping, but this information is not in the public domain and we were not granted access to it.

To collect the sample of reviews, we (a) searched the Scopus database with the string TITLE-ABS-KEY ( (furniture OR furnishing*) AND (flam* OR fire) ) AND (LIMIT-TO (DOCTYPE, "re") ), and (b) were provided with a list of documents of interest by BEIS/OPSS. The search results and list of documents were screened twice by one investigator (PW), with documents eligible for inclusion in the literature sample if they fulfilled the following conditions: they were about the involvement or behaviour of furniture in fires, or furniture as a source of exposure to fire retardants; were review articles; were in English; and were available in a format that could be imported into our document analysis environment (i.e. in electronic format and not protected by Digital Rights Management software).

In order to abstract concepts from our literature sample, we loaded the document set into the Atlas.ti computer-assisted qualitative data analysis software (CAQDAS) environment (Atlas.ti v9 for Windows). Documents were distributed among three annotators (NJ, CB, EW) who tagged the first occurrence of each term that potentially denoted a concept in furniture fire safety or flame retardant exposure. A second annotator (PW) checked each document for terms that may have been overlooked. To encourage consistency in tagging, annotators were trained in the use of Atlas.ti, and provided with annotation guidelines (S3 File). Regular checkpoint meetings were held to refine the annotation methodology, discuss preliminary findings and observations, suggest modifications to concept groups, and revise the annotation guidelines as needed. An excerpt of the annotation environment that shows document tags is illustrated in Fig 2. A flow-chart summarising the annotation process is presented on page 4 of S3 File.

Terms were then converted into concept networks in an interactive and reflective process involving the project team's designated experts in fire safety (RH) and CFR exposure (SH). Concepts were connected to each other in a concept network with four types of relationship: "is a", "is part of", "is a property of", and "affects". For example, [radiant heat]_is a_[ignition source], [lighter]_is a_[small open flame], [upholstery]_is part of_[combustible volume], [low relative body weight]_is a property of_[toddlers], and [intoxicated]_affects_[reactive capacity].

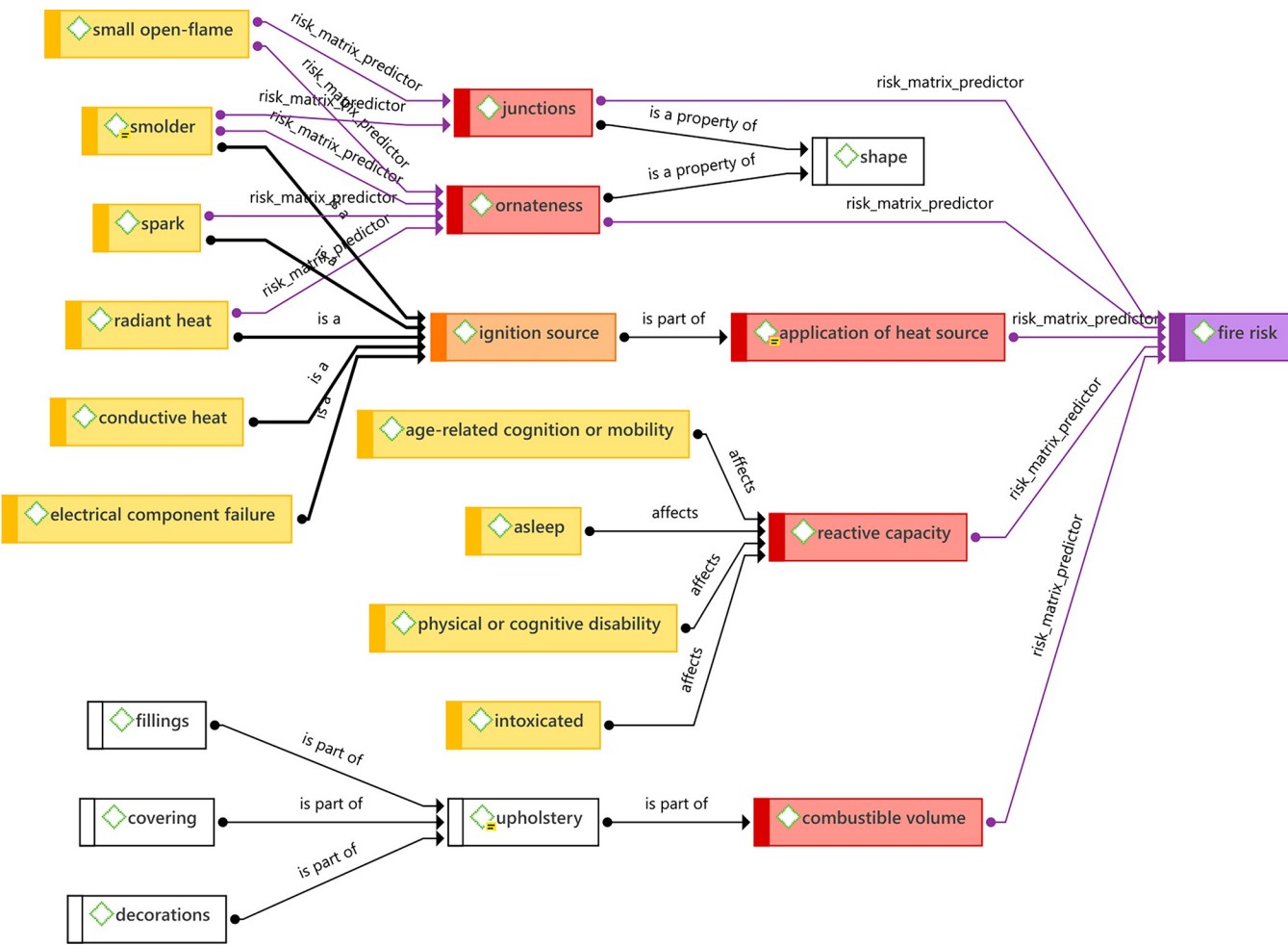

**Fig 2. High-level summary concept network for furniture fire risk.**

Duplicate or redundant terms were eliminated. Synonyms were clustered and a preferred term chosen to represent a concept. To increase the completeness of the concept networks, the designated experts could add concepts that had not been discovered from the literature sample. The process continued until a concept tree was assembled that sufficiently balanced simplicity, validity, and comprehensiveness for the purpose of defining furniture fire risk and CFR exposure models. The result of the process is not intended as a definitive model of how experts are discussing fire risk and CFR exposure in the literature; rather it is a minimal level of effort for identifying potential domains for the purpose of fire risk and CFR exposure modelling within the scope of this project.

**Results of search and concept mapping.** The Scopus search returned 111 results, of which 50 documents were determined to be eligible according to topic. Four results were not in English. 19 were not available as electronic documents, and 5 had Digital Rights Management protections. 8 documents were added from a list provided by BEIS-OPSS, plus the EN ISO 13943–2017 fire vocabulary standard. This gave a total of 31 documents (listed in S2 File). As a final additional document, we used the transcript of an interview we later conducted with two fire investigators about the role of furniture in domestic fires. The annotation process identified 2549 terms for concept mapping. After analysis, these terms were interpreted as representing 74 fire risk concepts and 36 CFR exposure concepts.

**Dimension selection: Fire risk model.** A high-level subsection of the fire risk concept network is shown in Fig 2. For the purposes of modelling fire risk of furniture while in use, a narrative emerges from the concept network. In general, fire risk is greater for furniture items that are in some combination (a) more likely to be used in proximity to an ignition source, (b) more likely to ignite as a result of exposure to an ignition source, (c) present a greater source of fuel, and (d) ignite when a user is in some way less able to react to a fire starting in the item. These can serve as four dimensions of fire risk (shown in Table 1).

For there to be any risk of fire, an ignition source needs to be present. These might be radiant (e.g. halogen heaters), conductive (e.g. electrical components), small open flames (e.g. cigarette lighters), or smouldering (e.g. cigarettes). A cigarette, lighter, or candle is an ignition source that can potentially initiate a fire in an item of furniture if it rolls into a junction between horizontally- and vertically-oriented surfaces or is caught in some decorative feature. For other heat sources, such as electrical components and halogen heaters, junctions are much less relevant as there is not an obvious sense in which a heater can be lodged in a junction, though decorative features (with protruding surfaces, tassels, etc.) may overheat and ignite or help a flame catch. Larger items pose higher risk because they provide more fuel for a fire. Items which people are more likely to be using in a state of reduced mobility (being asleep, intoxicated, with limitations on mobility due to physical or age- related disability) present higher risk because the user is less able to prevent injury by moving away from the ignited item.

**Selection of input data types: Fire risk model.** Input data to the fire risk model had to fulfil the following three conditions: that there is either sufficient empirical evidence or theoretical reason for the data category correlating with a model dimension (shown in Table 1); that the category of data can sufficiently differentiate fire risk between furniture product types within a model domain; and data for the category can be collected and analysed within the timeframe and resources available to the research team.

To estimate the amount of empirical evidence in the published literature that relates to each of the four risk model dimensions, we conducted a search for furniture fire safety literature in the Scopus database. This search yielded 3885 documents (for the search string and full list of documents, see S11 File). We then created a thesaurus of fire risk terms and counted the frequency of occurrence of these terms in the titles and abstracts of documents from our search, to derive an estimate of how much evidence there is in relation to each concept in our concept model, and therefore for each domain in our risk model.

The results of this exercise suggests that empirical research into furniture fires most frequently focuses on ignition tests on furniture and fabric (domain #2), and heat release from flame spread through items of furniture or cushioning and fabric materials (#3). Otherwise, there seems to be very little published research into how user behaviour, furniture form factors (e.g. shape), or user vulnerability affects fire risk in upholstered furniture (relevant data are in S10 File, in the worksheets "Code Thesaurus" and "Risk Term Frequency in Texts"). In our experience, fire statistics do not have sufficient resolution to differentiate the relative contribution of specific furniture types to house fires.

**Data collection: Furniture fire risk model.** We selected five data categories for our furniture fire risk model: likelihood of contact of furniture type with an ignition source; number of junctions in the type; ornateness of the type; combustible volume of the type; and likely reactive capacity of the user of the type of furniture. Model dimensions, data categories, and input data are summarised in Fig 5.

Data collection proceeded as follows. Evaluators were presented with a questionnaire for 30 furniture product types. The furniture types were selected by the research team (NB) in consultation with furniture manufacturers as small items of furniture likely to be within scope of the

**Table 1. Reasons for selection of data categories to represent fire risk model dimensions.**

| Model dimension | Potential correlates | Sufficiently differentiates between furniture types? | Measurable with available resources? | Included in the model? |
|---|---|---|---|---|
| Exposure to a source of ignition | Likelihood of contact with source of ignition: Higher likelihood of contact with any source of ignition should correlate with higher fire risk | Yes: lit cigarettes or candles more likely to be dropped on sofas or armchairs than cribs or high chairs; halogen heaters more likely to be used near armchairs than near pushchairs | Yes: while empirical data on contact with ignition sources is not readily available, an informed person should be able judge relative likelihood of contact of an item of furniture with an ignition source, given how an item of furniture is generally used | Yes |
| Potential for ignition of item of furniture | Number of junctions: Joins (junctions) between horizontal and vertical surfaces are where fires start from dropped cigarettes and candles, so more junctions should correlate with higher risk [2]. Primarily relevant for dropped smokers materials. | Yes: dining chairs usually have one junction; armchairs 3 junctions (see Fig 3), and bassinets 4 junctions (effectively an open box) | Yes: junctions can be counted | Yes |
| | Ornateness: More complex surfaces offer more catch-points for ignition and spread of flames, so should to some extent correlate with fire risk | Yes: playmats tend to be plainer objects with few exposed corners, while cushions tend to be more ornate with at least four exposed corners (see Fig 4 for an illustration of relative ornateness) | Yes: an informed person should be able to make a reasonably reliable judgement about relative ornateness | Yes |
| | Flammability of covering: A flammable covering provides more opportunity for ignition, correlating with higher risk | Unclear: it is not clear if there are systematic differences in choice of cover materials for different furniture types. Some may be more likely to be wipe-clean (e.g. infant products), but the relationship between this and flammability is unclear | No: data on fabric flammability is available, but data on the coverings used in furniture types is not sufficiently accessible given available resources | No |
| | Flammability of filling: Flammable filling provides more opportunity for a fire to take hold and spread, correlating with higher risk | Unclear: it is not clear if there are systematic differences in choice of filling materials for different furniture types | No: while data on flammability of fillings is available, data on the fillings used in different furniture types is not sufficiently accessible given available resources | No |
| | Flammability of fabric / filling combination: Arguably the strongest correlate to fire risk, insofar as the material composition of an item of furniture impacts fire risk | Unclear: it is not clear if there are systematic differences in fabric / filling combinations for different furniture types | No: sufficient data on flammability of fabric / filling combinations is probably not available; data on the fabric / filling combinations used in different furniture types is not sufficiently accessible given available resources | No |
| Potential for spread of flame through the item of furniture | Higher combustible volume: Larger items of furniture provide more fuel and therefore potential for fire to grow | Yes: some types of furniture are larger than others (e.g. sofas vs. high chairs) | Yes: the volume of an item of furniture can be calculated | Yes |
| | Heat release from fabric / filling combination: As a measure of speed and amount of energy that can be released by an item, heat release provides arguably the strongest correlate to fire risk, insofar as the material composition of an item of furniture impacts fire risk | Unclear: it is not clear if there are systematic differences in fabric / filling combinations for different furniture types | No: sufficient data on flammability of fabric / filling combinations is probably not available; data on the fillings used in different furniture types is not sufficiently accessible given available resources | No |
| Vulnerability of nearby person to fire in item | Reactive capacity of person: Unconsciousness through sleep or intoxication, physical or cognitive disabilities, whether through age (i.e. very old or very young), or other factors, all potentially restrict a person's capacity to detect and react to fire and therefore correlates with increased risk of injury or death | Yes: armchairs and cots are more likely to be used for sleeping than dining chairs or high chairs | Yes: an informed person should be able make reasonably reliable judgements about the reactive capacity of a user of an item of furniture, given how the item is generally used | Yes |

new FFRs. For each type, an example item of relatively high fire risk and relatively low fire risk was shown (also selected by NB), with the evaluators completing the questionnaire for both items. This allowed us to put an approximate higher and lower range on evaluations of furniture product types.

We used informed judgement to estimate on a scale of 1–5 likelihood of contact with an ignition source, ornateness of surface and shape, and reactive capacity of a user or nearby person. A score of 1 means the evaluator considered the item to be as little ornate, unlikely to be used by someone with restricted reactive capacity, or unlikely to come into contact with a source of ignition, as they could conceive. Scores of 5 meant the opposite. We counted the number of junctions. For combustible volume we estimated the volume in cubic metres ($m^3$) of the upholstered elements of an item of furniture. Evaluators were provided with guidance notes to facilitate comprehensive and consistent consideration of risk concepts that related to each data category.

Three evaluators (CB, NB, EP) independently completed all the questions in each questionnaire in random order to minimise learning effects, before convening in a consolidation session (moderated by PW) where they came to consensus on each score. Finally, an external set of evaluators were asked to complete a questionnaire for a subset of four furniture product types, to indicate the extent to which the research team evaluations generalise to evaluations made by external experts. Hypothetical evaluations of two items of furniture are shown for illustration in Fig 6. Evaluation questionnaires are shown in S12 File. Calculations of volume are shown in S13 File.

**Calculation of model output: Furniture fire risk model.**   We used two transformation functions (marked as T in Fig 5) to convert junctions and combustible volume to a scale of 1–5. This was to render them combinable with the Likert scores and reflect empirical uncertainty about precisely how much each of these factors contributes to overall furniture fire risk. No transformations were applied to the Likert scores.

Junction counts were transformed such that zero junctions scored 1, one scored 2, two scored 3, three scored 4, and four or more scored 5. This was on the rationale that an object with four junctions (such as a basket or bassinet) would effectively be an open box, from which an ignition source would not be able to roll anywhere except into a junction, and therefore represents maximum risk in that domain (see Fig 3 for an illustration of junction counts). Objects with 5 or more junctions were considered complex objects and assumed to present equal risk to an open box. Because junctions only present a fire risk in relation to ignition sources that can fall into a junction point in an item of furniture, the junction score was only included in the overall risk score if the evaluators referred to candles or smokers materials in the narrative justification for scoring ignition source. Otherwise, the junction score would be set to 0. (Narrative justifications are shown in the "Furniture Justifications" sheet of S16 File).

Combustible volume was assumed to have linear increase in risk up to a maximum score of 5 for a volume of 0.009 $m^3$. 0.009 $m^3$ is the volume of a small cushion, a threshold above which we assumed that post-ignition flame spread would effectively be guaranteed. The transformation functions are shown in the "Risk Tables" sheet of S16 File.

For calculating the model output, we assumed that fire risk follows a failure model, whereby exposure to an ignition source has to precede ignition, that has to precede fire spread, that has to occur when a user is unable to react, in order for injury to follow. We decided not to make risk equal to the lowest score (a pure clamp-point model, where the risk of the outcome is determined by the strongest link in the failure chain) as we felt this was too deterministic in an environment where we did not have high certainty in the data going into the model, and we were assuming no interactions between the dimensions of the model. Instead, we calculated

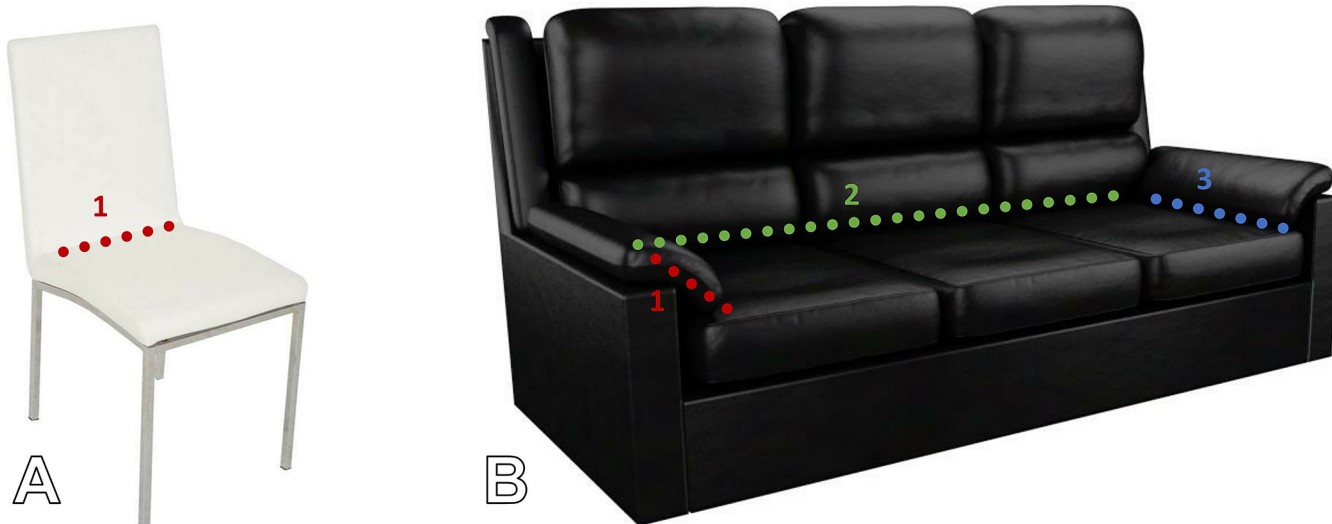

**Fig 3. Illustration of junction counts in two items of furniture.** Image of chair (A): Just Dining Chairs, Flickr, reproduced under a Creative Commons BY 2.0 licence. Image of sofa (B): Monkeywing, Flickr, reproduced under a Creative Commons BY 2.0 licence.

the lowest quartile to hedge on the lowest scores, but not have the single lowest score determining the model output.

Risk scores for the hypothetical examples (Fig 6) suggest that a plain dining chair presents lower fire risk than a large sofa. While the scores are relative rather than absolute, and based on indirect measures that imperfectly correlate with fire risk, some of which are subjective human judgements, the model does seem to produce an intuitively meaningful result via a simple and transparent process.

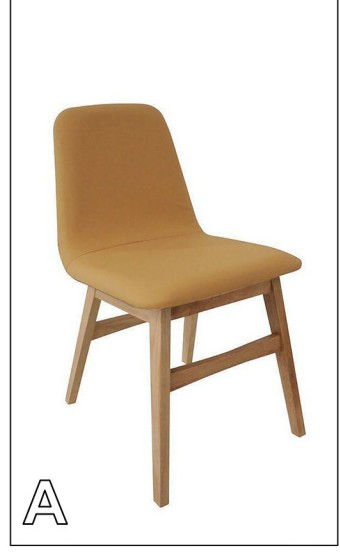
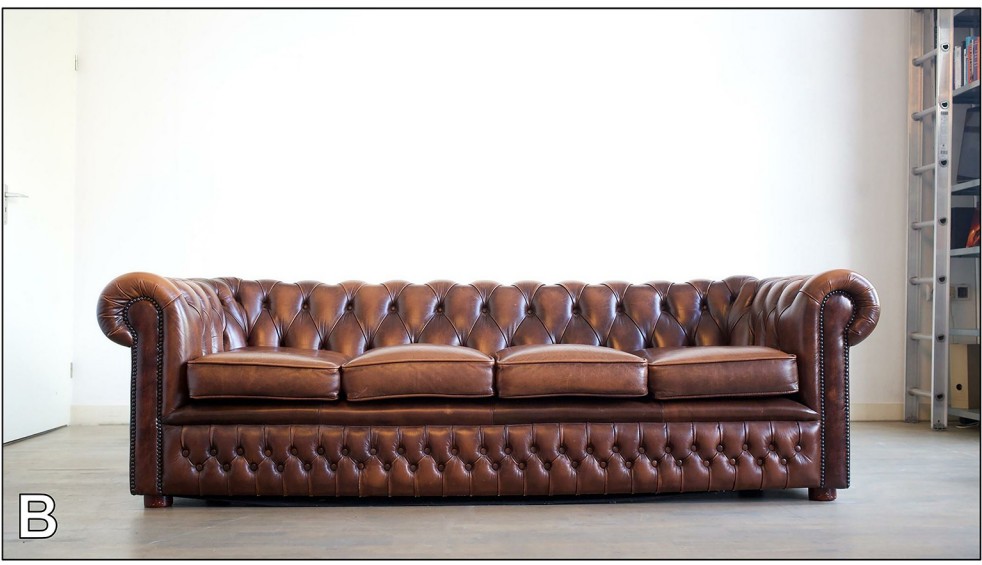

**Fig 4.** Illustration of a relatively less ornate chair (A) and more ornate sofa (B). Image of chair: Just Dining Chairs, Flickr, reproduced under a Creative Commons BY 2.0 licence. Image of sofa: Sebastian ter Burg, Flickr, reproduced under a Creative Commons BY 2.0 licence. (Images also used in Figs 6 and 9).

Because there are two termini (injury and property damage) to our risk model, we calculated two model outputs, showing injury risk as a function of all four domains as our primary output, and damage risk as a function of all domains except reactive capacity as a secondary output. Since property damage tracked risk of injury, and risk of injury allows vulnerability of person to be used as data in the model, for the model output we prioritised risk of injury.

**Dimension selection: CFR exposure.** A high-level subsection of the CFR exposure concept network is shown in Fig 7. The full network is shown in S7 File. For the purposes of modelling potential for CFR exposure from furniture while in use, a narrative emerges from the concept network. In general, exposure to CFRs from furniture is from two direct routes and one indirect route, with two major factors modifying exposure. The direct routes are dermal migration (skin contact), oral migration (contact via mouthing behaviours), and the indirect route is release of CFRs from an item to the indoor environment (e.g. via dust, off-gassing, abrasion, etc.). The two modifying factors are duration of contact (the more an item is used, the greater potential for exposure), and low relative body weight of the user (higher surface area to volume ratio leads to greater relative exposure).

**Selection of input data types: CFR exposure model.** Input data into the CFR exposure model had to fulfil three conditions: that there is either good empirical evidence or good theoretical reasons for the data category correlating with a model domain; that the category of data can differentiate potential for CFR exposure between furniture product types within a model domain; and the data category is accessible to informed judgement or empirical measurement. Reasons for selection of data categories for the CFR exposure model are summarised in Table 2.

It was not feasible to conduct the same keyword analysis of the literature as we did for fire risk to evaluate the potential availability of empirical data relating to exposure, as the literature on exposure was too extensive to be searched, screened, and analysed within the six month timeframe of the project. We do, however, know that none of the reviews in our document set that addressed human exposure to flame retardants discussed in detail how specific items of furniture contribute to a person's exposure to CFRs. We also know that exposure studies are not able to differentiate exposure sources to that degree of granularity. While it is possible, for example, to measure how much people are exposed to CFRs in the immediate domestic environment, we are not aware of any published data showing how much CFR load is specifically from furniture, especially from individual furniture product types. (This is explained in more detail in S17 File). Similar to characterising fire risk, this left us in a situation whereby, although we have a comprehensive list of factors that could affect CFR exposure from furniture, there is a lack of empirical data that can quantify how much each risk dimension contributes to overall exposure risk when considering furniture product types.

**Data collection: CFR exposure.** We used the same overall method for collecting data for modelling CFR exposure potential as we did for the furniture fire risk model, with a questionnaire including term definitions and prompts for 30 furniture types and the same examples of low- and high-risk furniture (S14 and S15 Files). We used informed judgement to score on a scale of 1–5 likelihood of use of an item by a young child, likelihood of contact with bare skin, likelihood of mouthing, and cumulative use of an item. Surface area was estimated in square metres ($m^2$), based on the shape and reported dimensions of an item of furniture. Two evaluators (PW and SH) completed each questionnaire before discussing and reaching consensus on scores. External validation for exposure questionnaires was not conducted. This was for capacity reasons (the scores are less controversial and more certain, so capacity for validation was reserved for the fire risk model). Model dimensions, data categories, and input data for the CFR exposure model are summarised in Fig 8.

**Table 2. Reasons for selection of data categories to represent CFR exposure model dimensions.**

| Model domain | Potential correlates | Sufficiently differentiates between furniture types? | Measurable with available resources? | Included in the model? |
|---|---|---|---|---|
| Low relative body weight of user | Likelihood of use of item by young child: young children have a much higher surface area to volume ratio than adults, therefore likelihood of use of an item of furniture by a young child should correlate with low relative body weight of a user [20]. | Yes: Some furniture (e.g. cot mattress) is specifically designed for infants and toddlers to use; therefore, potential for exposure according to likelihood of use by a young child should differentiate between furniture product types. | Yes: It seems reasonable that an informed evaluator can make a reasonably reliable judgement about when an item of furniture is likely to be used by an infant or toddler. | Yes |
| Dermal migration | Likelihood of contact with bare skin: Dermal migration is reduced by clothing, so increased dermal migration should correlate with increased likelihood of direct skin contact with an item of furniture [21]. | Yes: Some types of furniture are more likely to be used relatively undressed or be in contact with bare skin than others (pillows will contact bare heads, whereas, for example, dining chairs will be less likely to be in contact with bare skin). | Yes: It seems reasonable that an evaluator could judge the amount of bare skin likely to be in direct contact with an item of furniture. | Yes |
| Oral migration | Likelihood of mouthing of the item: Direct oral contact with an item of furniture will increase the potential for CFR exposure [22]. | Yes: Since mouthing behaviours are specific to very young children, and some furniture is designed for young children, mouthing behaviour should differentiate furniture types. | Yes: It should be possible for an informed evaluator to judge the relative likelihood that a young child could mouth a given item of furniture. | Yes |
| Duration of contact | Cumulative time of use of the item: The more frequently and the longer the period of time for which someone uses an item of furniture, the greater the exposure to CFRs should be via whatever route. More heavily-used items can also be expected to release more CFRs abrasion and, if warmed by body temperature, also through volatilisation [23]. | Yes: Some furniture product types are also designed to be used for longer periods of time than others (e.g. beds or armchairs vs. dining chairs) | Yes: It should be possible for an informed evaluator to judge the relative duration of contact with an item of furniture. | Yes |
| Size of item | Surface area of the item: Larger items are a larger potential reserve of CFRs and can therefore be expected to release more fire retardants; since surface area correlates with size, and likely correlates better with CFR release than volume alone, it seems reasonable to use surface area as the correlate for size of item [24, 25]. | Yes: Since some types of furniture differ in size (e.g. adult beds are larger than child beds, and armchairs are larger than cushions), surface area should differentiate between furniture product types. | Yes: The surface area of an item can be calculated from information about the product dimensions. | Yes |

**Calculation of model output: CFR exposure.** We used a log transformation function to convert surface area to a scale of 1–5. This was to render the measure combinable with the Likert scores, reflect empirical uncertainty about how much each of these factors actually contributes to CFR exposure, and enable small changes in surface area for small objects to be as important as large changes in surface area for large objects. There is a risk that the log transform may underestimate exposure from large items; to allow reproducibility and third-party testing of our assumptions, the log transform is provided in the Risk Tables sheet of S16 File.

We used the arithmetic mean for calculating overall CFR exposure scores. This was to reflect a lack of rationale for weighting one domain more heavily than any other. Because the judgements from which the scores are derived are based on perceived relative importance, rather than being grounded in empirical evidence of absolute exposure to CFRs, the model shows relative potential for exposure between furniture product types.

Risk scores for the hypothetical examples (Fig 9) suggest that a plain dining chair presents lower potential for CFR exposure than a large sofa. As for the fire risk model, while the scores are relative rather than absolute and based on indirect measures that imperfectly correlate with fire risk, some of which are subjective human judgements, the model does seem to produce an intuitively meaningful result via a simple and transparent process.

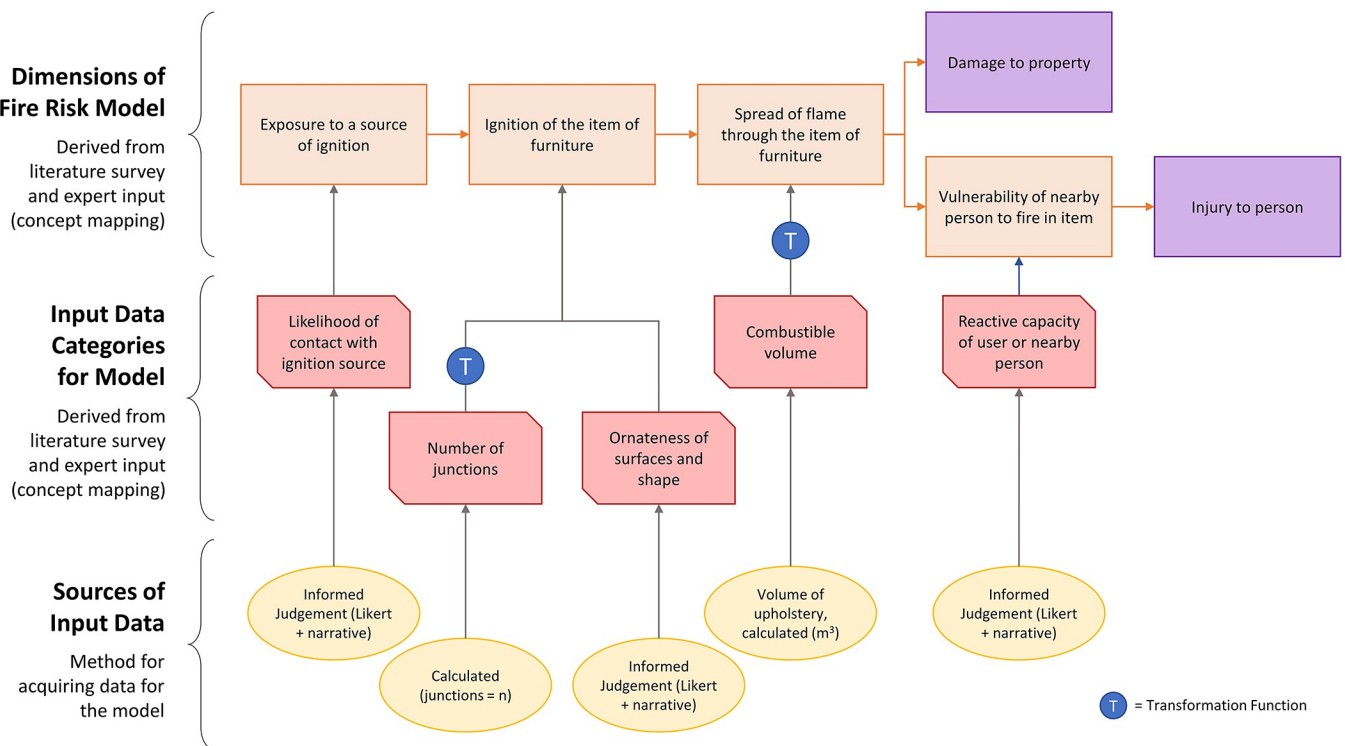

**Fig 5. Furniture fire risk model with input data categories and data sources.** Colours match the network diagrams (S4–S9 Files) to facilitate cross-checking of how the model is derived from the concept network.

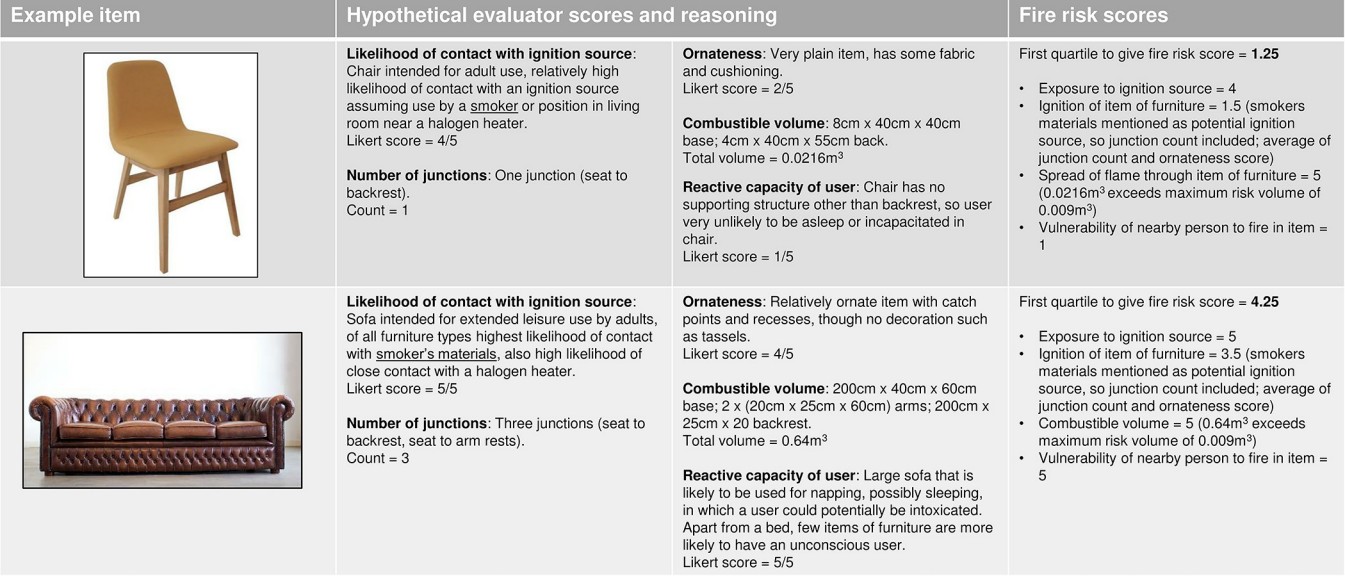

| Example item | Hypothetical evaluator scores and reasoning | | Fire risk scores |
|---|---|---|---|
| | **Likelihood of contact with ignition source**: Chair intended for adult use, relatively high likelihood of contact with an ignition source assuming use by a <u>smoker</u> or position in living room near a halogen heater. Likert score = 4/5 <br><br>**Number of junctions**: One junction (seat to backrest). Count = 1 | **Ornateness**: Very plain item, has some fabric and cushioning. Likert score = 2/5 <br><br>**Combustible volume**: 8cm x 40cm x 40cm base; 4cm x 40cm x 55cm back. Total volume = 0.0216m³ <br><br>**Reactive capacity of user**: Chair has no supporting structure other than backrest, so user very unlikely to be asleep or incapacitated in chair. Likert score = 1/5 | First quartile to give fire risk score = **1.25** <br><br>• Exposure to ignition source = 4 <br>• Ignition of item of furniture = 1.5 (smokers materials mentioned as potential ignition source, so junction count included; average of junction count and ornateness score) <br>• Spread of flame through item of furniture = 5 (0.0216m³ exceeds maximum risk volume of 0.009m³) <br>• Vulnerability of nearby person to fire in item = 1 |
| | **Likelihood of contact with ignition source**: Sofa intended for extended leisure use by adults, of all furniture types highest likelihood of contact with <u>smoker's materials</u>, also high likelihood of close contact with a halogen heater. Likert score = 5/5 <br><br>**Number of junctions**: Three junctions (seat to backrest, seat to arm rests). Count = 3 | **Ornateness**: Relatively ornate item with catch points and recesses, though no decoration such as tassels. Likert score = 4/5 <br><br>**Combustible volume**: 200cm x 40cm x 60cm base; 2 x (20cm x 25cm x 60cm) arms; 200cm x 25cm x 20 backrest. Total volume = 0.64m³ <br><br>**Reactive capacity of user**: Large sofa that is likely to be used for napping, possibly sleeping, in which a user could potentially be intoxicated. Apart from a bed, few items of furniture are more likely to have an unconscious user. Likert score = 5/5 | First quartile to give fire risk score = **4.25** <br><br>• Exposure to ignition source = 5 <br>• Ignition of item of furniture = 3.5 (smokers materials mentioned as potential ignition source, so junction count included; average of junction count and ornateness score) <br>• Combustible volume = 5 (0.64m³ exceeds maximum risk volume of 0.009m³) <br>• Vulnerability of nearby person to fire in item = 5 |

**Fig 6. Hypothetical examples to illustrate calculation of fire risk scores.** Two example items, the small chair and large sofa from Fig 4, are shown. Hypothetical evaluator scores and reasons are given for each input data category (see Fig 5). The translation into dimension scores and calculation of overall fire risk score is then given.

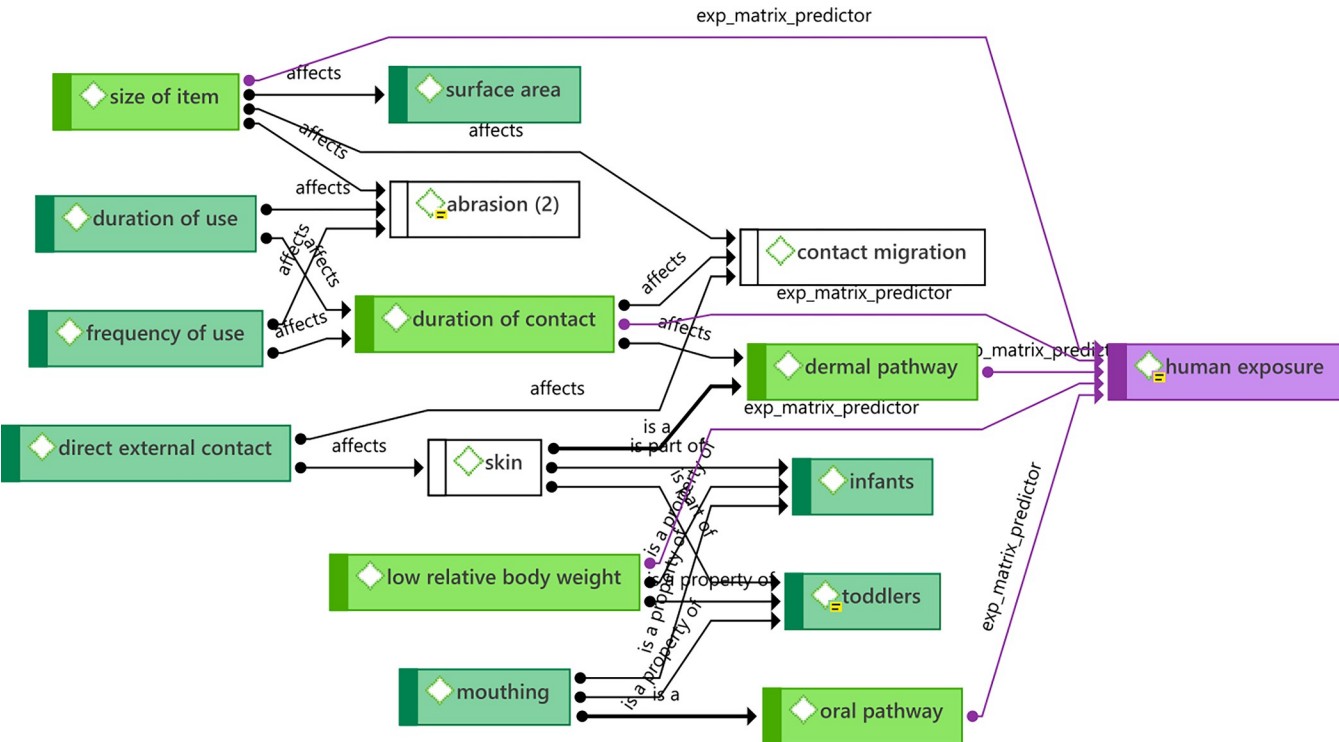

**Fig 7. High-level summary concept network for potential for CFR exposure.**

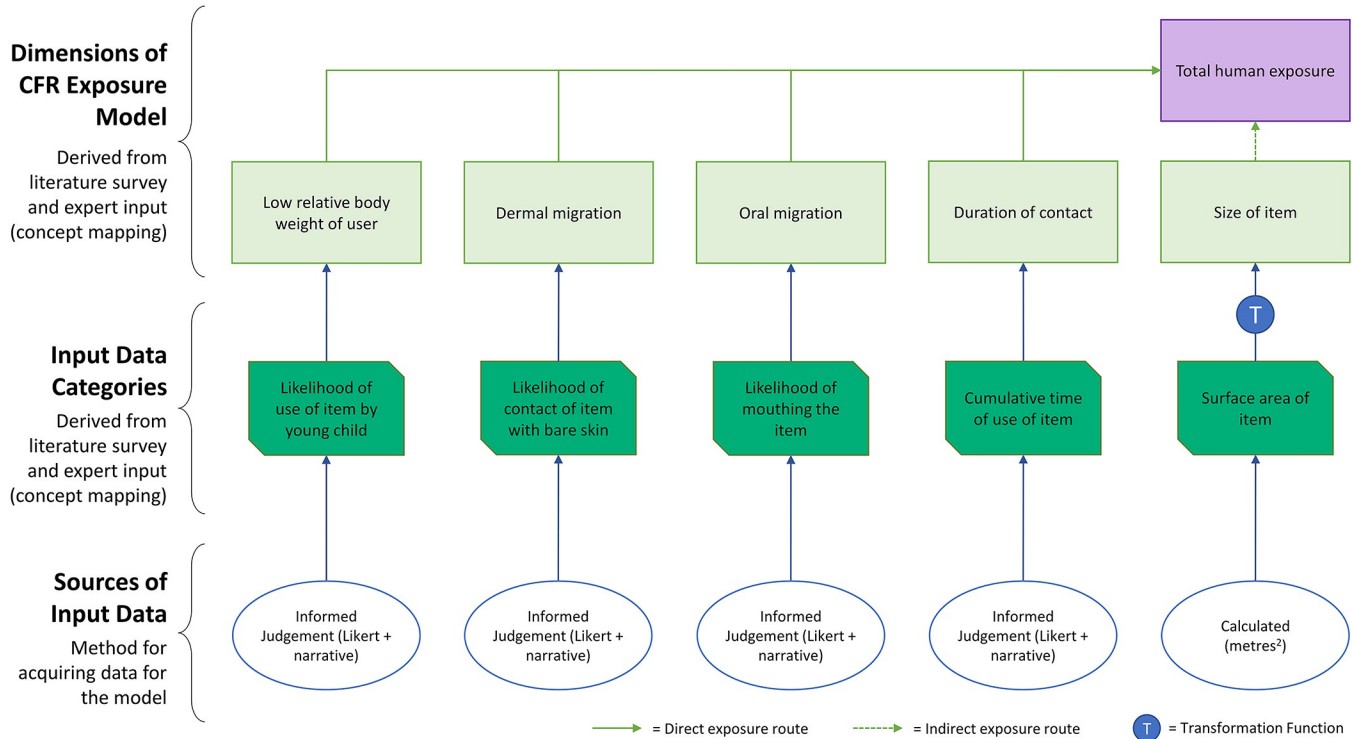

**Fig 8. CFR exposure model with input data categories and data sources.** Colours match the network diagrams (S4–S9 Files) to facilitate cross-checking of how the model is derived from the concept network.

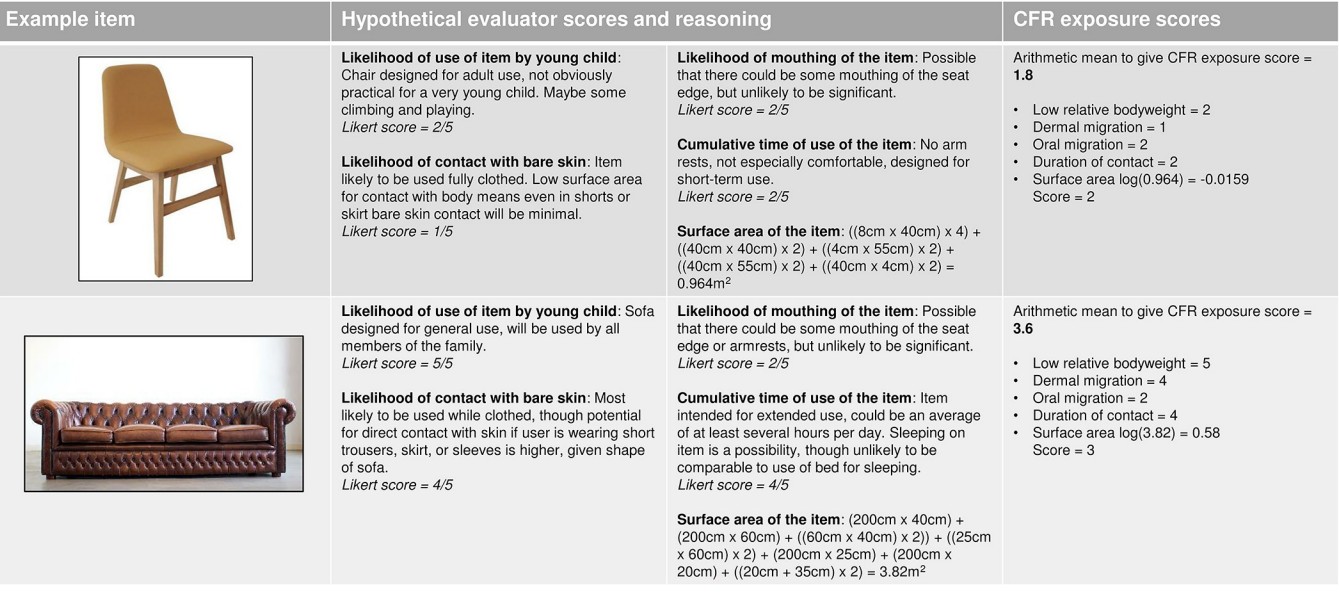

| Example item | Hypothetical evaluator scores and reasoning | | CFR exposure scores |
|---|---|---|---|
| | **Likelihood of use of item by young child**: Chair designed for adult use, not obviously practical for a very young child. Maybe some climbing and playing. *Likert score = 2/5*<br><br>**Likelihood of contact with bare skin**: Item likely to be used fully clothed. Low surface area for contact with body means even in shorts or skirt bare skin contact will be minimal. *Likert score = 1/5* | **Likelihood of mouthing of the item**: Possible that there could be some mouthing of the seat edge, but unlikely to be significant. *Likert score = 2/5*<br><br>**Cumulative time of use of the item**: No arm rests, not especially comfortable, designed for short-term use. *Likert score = 2/5*<br><br>**Surface area of the item**: ((8cm x 40cm) x 4) + ((40cm x 40cm) x 2) + ((4cm x 55cm) x 2) + ((40cm x 55cm) x 2) + ((40cm x 4cm) x 2) = $0.964m^2$ | Arithmetic mean to give CFR exposure score = **1.8**<br><br>• Low relative bodyweight = 2<br>• Dermal migration = 1<br>• Oral migration = 2<br>• Duration of contact = 2<br>• Surface area log(0.964) = -0.0159<br>  Score = 2 |
| | **Likelihood of use of item by young child**: Sofa designed for general use, will be used by all members of the family. *Likert score = 5/5*<br><br>**Likelihood of contact with bare skin**: Most likely to be used while clothed, though potential for direct contact with skin if user is wearing short trousers, skirt, or sleeves is higher, given shape of sofa. *Likert score = 4/5* | **Likelihood of mouthing of the item**: Possible that there could be some mouthing of the seat edge or armrests, but unlikely to be significant. *Likert score = 2/5*<br><br>**Cumulative time of use of the item**: Item intended for extended use, could be an average of at least several hours per day. Sleeping on item is a possibility, though unlikely to be comparable to use of bed for sleeping. *Likert score = 4/5*<br><br>**Surface area of the item**: (200cm x 40cm) + (200cm x 60cm) + ((60cm x 40cm) x 2)) + ((25cm x 60cm) x 2) + (200cm x 25cm) + (200cm x 20cm) + ((20cm x 35cm) x 2) = $3.82m^2$ | Arithmetic mean to give CFR exposure score = **3.6**<br><br>• Low relative bodyweight = 5<br>• Dermal migration = 4<br>• Oral migration = 2<br>• Duration of contact = 4<br>• Surface area log(3.82) = 0.58<br>  Score = 3 |

**Fig 9. Hypothetical examples to illustrate calculation of CFR exposure scores.** Two example items, the small chair and large sofa from Fig 4, are shown. Hypothetical evaluator scores and reasons are given for each input data category (see Fig 8). The translation into dimension scores and calculation of overall exposure score is then given.

## Fire risk and CFR exposure reconciliation

**Ranking and clustering furniture types.** We ranked furniture types and identified product clusters using three methods. First, we calculated their relative position in a two-dimensional space, plotting the mean fire risk score against the mean CFR exposure score for the higher- and lower-risk furniture items for each type. We plotted the higher and lower range (i.e. the minimum and maximum) for each furniture type as an ellipse around the mean score. Second, we calculated the Euclidean distance in the space defined by furniture injury risk, damage risk, and CFR exposure scores, and used a dendrogram to visualise the resulting clusters, a process facilitated by examining links that cross a given horizontal point in the dendrogram. Thirdly, on the scatter plot we marked relative thresholds for injury risk on the perpendicular to the X axis and CFR exposure on the Y axis to determine whether a quadrant might capture a coherent group of furniture types.

Dendrograms facilitate visualisation of groups by recalculating arrays of two or more dimensions as a one-dimensional distance in Euclidean space. Items clustering insofar as they occupy a similar region in that space. The top of each u-shaped link between furniture product types shows the distance between items. Taller cross bars indicate larger differences. Drawing a horizontal line across the dendrogram indicates clusters of items, insofar as two or more items are joined by a u-shaped link below the horizontal line. For the dendrogram, mean injury risk, damage risk, and exposure potential scores for each furniture product type were recalculated as a one-dimensional distance in Euclidean space.

**Sensitivity testing.** A significant proportion of the data being used for the FFS matrix is based on subjective judgement. Because such judgements may vary between individuals, and could vary systematically between different groups of evaluators, it was important to test the sensitivity of the results of the risk matrix to variation in evaluator judgement. The critical domain for sensitivity analysis was likelihood of contact of a furniture product type with an ignition source. Where other domains involved readily-observable events and behaviours (e.g.

likelihood of mouthing, potential for a user being asleep), fire service professionals may have a more accurate and/or systematically different perception of likelihood of contact of an item of furniture with an ignition source to the evaluators in the research team.

Since the objective of the matrix is to identify clusters of furniture types, our approach to sensitivity testing was to determine the extent to which differences in evaluator responses threatened the ability of our approach to differentiate such clusters. To do this, we first modelled the effect that systematic error in the contact domain would have on overall fire risk scores. Here, we assumed that contact scores for each evaluated item (n = 60, with a high and low variant for each furniture product type) were underestimated by 1 point, 2 points, etc., capping the maximum error at a score of 5. We plotted all the injury scores for each item of furniture against imputed error, adding a small amount of jitter to each point so relative position could be seen on the plot. The point at which clusters disappeared was the point at which the fire risk model could no longer differentiate relative fire risk between furniture types. We then asked members of the project advisory panel with fire investigation experience (n = 3) to complete an evaluation questionnaire for a subsample (n = 4) of the furniture product types (armchairs, pushchairs, headboards, and pet beds), to determine if their answers were within the tolerances indicated by our sensitivity test.

## Results

### Fire risk and CFR exposure scores

Table 3 shows the fire risk and CFR exposure model scores for the low-risk variant, high-risk variant, and the mean of the two variants for each product type included in our study. Fig 10 shows the rank ordering of product types by mean fire risk score, and the range for high and low risk variants within each type. Fig 11 shows the same, but for CFR exposure.

14 of the product types are intended for use by small children (infants and very young children). Bassinets, play mats, carry cots, and baby nests score for fire risk in a range approximately comparable to armchairs. Car seats, baby mattresses, prams, and play pens score among the lowest furniture types for fire risk. For CFR exposure potential, 12 of the 14 small child product types are in the top 15 scores. Exposure scores for small child products have a more distinctive distribution than fire risk scores, with arguably only bean bags as a product type showing an exposure score range approximately comparable to most small child products.

### Furniture clusters

The dendrogram is suggestive of five major product groupings (Fig 12 and Table 4). Baby changing mats, baby mattresses, prams, and playpens are a distinct group of their own. A second group includes small child products such as side rails and car seats, along with floor cushions and bean bags. A third group includes several small child products including bassinets, carry cots, play mats, and light-up cushions. Other product groupings include bed parts and armchairs, and pet beds and padded foot stools. These groupings make a certain amount of intuitive sense. For example, play mats and bassinets have similar fire risk and exposure profiles, and are differentiated from carry cots due to the impact of their lack of portability on the likelihood of exposure to a source of ignition. Similarly, armchairs and upholstered bed bases form an outlying cluster due to their size and being part of a relatively small group of furniture types in which a person can be expected to be asleep and in direct contact for extended periods of time. The suggestion of groups and trends is also suggested by the scatter plot, with a tendency for small child products to gravitate towards the upper-left higher exposure, lower fire risk quadrant (Fig 13).

**Table 3. Fire risk (injury) and CFR exposure scores.**

| Furniture Type | Risk of Injury Score | | | CFR Exposure Potential | | |
|---|---|---|---|---|---|---|
| | Low | High | High-Low Mean | Low | High | High-Low Mean |
| Armchairs | 2.00 | 4.63 | 3.31 | 1.80 | 3.00 | 2.40 |
| Baby changing mat | 1.63 | 1.75 | 1.69 | 3.00 | 3.20 | 3.10 |
| Baby mattress | 1.63 | 1.75 | 1.69 | 3.20 | 3.40 | 3.30 |
| Baby nests | 3.13 | 3.13 | 3.13 | 3.20 | 3.40 | 3.30 |
| Baby products with seat | 2.38 | 3.38 | 2.88 | 3.40 | 3.60 | 3.50 |
| Bassinet | 2.38 | 2.63 | 2.50 | 3.20 | 4.00 | 3.60 |
| Bean bag | 1.63 | 3.38 | 2.50 | 2.40 | 3.40 | 2.90 |
| Car seat | 1.88 | 2.00 | 1.94 | 3.40 | 3.40 | 3.40 |
| Carry cots | 3.13 | 3.50 | 3.31 | 3.40 | 3.60 | 3.50 |
| Child trailers and strollers | 1.00 | 2.50 | 1.75 | 3.00 | 3.60 | 3.30 |
| Divan | 1.75 | 1.75 | 1.75 | 1.60 | 1.60 | 1.60 |
| Floor cushion | 2.00 | 1.75 | 1.88 | 3.40 | 3.20 | 3.30 |
| Footboard | 2.38 | 2.63 | 2.50 | 1.80 | 2.00 | 1.90 |
| Headboard | 2.63 | 3.88 | 3.25 | 2.00 | 2.40 | 2.20 |
| Light-up children's cushion | 2.75 | 2.75 | 2.75 | 3.60 | 3.60 | 3.60 |
| Living aids | 2.38 | 3.63 | 3.00 | 2.00 | 3.20 | 2.60 |
| Loose and stretch covers | 2.75 | 4.00 | 3.38 | 2.80 | 3.80 | 3.30 |
| Outdoor furniture | 1.38 | 2.75 | 2.06 | 1.60 | 3.40 | 2.50 |
| Outdoor furniture separate upholstery | 1.88 | 2.75 | 2.31 | 2.60 | 2.80 | 2.70 |
| Padded foot stools | 2.63 | 2.75 | 2.69 | 1.80 | 2.40 | 2.10 |
| Pet beds | 1.75 | 3.00 | 2.38 | 1.60 | 2.20 | 1.90 |
| Pillow | 2.00 | 2.88 | 2.44 | 4.20 | 3.20 | 3.70 |
| Play mat | 1.75 | 3.50 | 2.63 | 3.20 | 3.60 | 3.40 |
| Playpens | 1.88 | 1.75 | 1.81 | 3.80 | 4.00 | 3.90 |
| Prams | 1.38 | 1.75 | 1.56 | 3.40 | 3.60 | 3.50 |
| Scatter cushion | 2.75 | 3.00 | 2.88 | 3.00 | 3.20 | 3.10 |
| Seat pad | 2.00 | 2.88 | 2.44 | 2.40 | 2.80 | 2.60 |
| Separate baby upholstery | 2.50 | 1.75 | 2.13 | 3.20 | 3.20 | 3.20 |
| Side rails | 1.38 | 3.00 | 2.19 | 3.40 | 2.20 | 2.80 |
| Upholstered bed base | 2.75 | 3.75 | 3.25 | 2.00 | 3.00 | 2.50 |

## Sensitivity testing

The sensitivity plot is shown in Fig 14 (see "Fire Contact Sensitivity" sheet of S22 File for source file). The plot shows that, even with a simulated 2-point difference in judgement between our evaluators and a hypothetical external group of evaluators, furniture types still appear to cluster (there are distinct clusters in the plotted vertical distributions for 1- and 2-point differences; the clusters become less distinct for point differences of 3 or more).

Project advisory panel evaluations are shown in Table 5. For the contact domain, in general the research team ("Team, Consolidated" in the table) were more conservative in their judgements than the external evaluators, and the difference between the evaluations was within range of the tolerances indicated by the sensitivity analysis. For other domains, judgements often differed significantly; however, the reasons for this were often to do with incorrect interpretation of a question (e.g. basing evaluations of ornateness on constituent materials) and sometimes internally inconsistent (e.g. scoring an armchair differently to a headboard, while stating that both could be used by a person who is asleep and has physical disabilities). Such

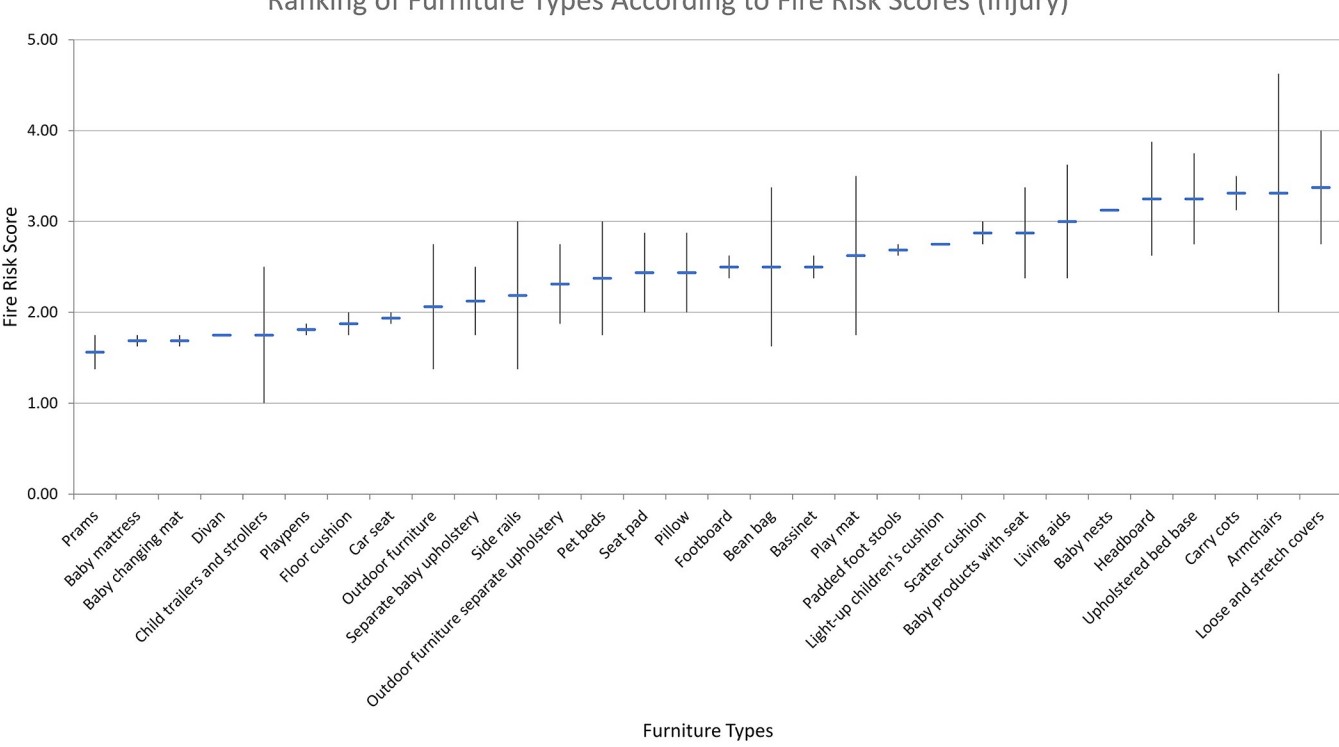

**Fig 10. Ranking of furniture types according to fire risk score (injury).**

issues are likely the result of the external evaluators not going through the same training or consolidation process that the research team followed, which helps secure agreement on how to interpret a question and improves the consistency of evaluation.

## Discussion

Below we discuss the implications of our work for understanding the role of furniture in fires, draw out some of the strengths and limitations of our approach, caution against over-interpretation of our data and models, and make recommendations for how the models and matrix might be further developed.

### Product groups

While identifying clusters is an interpretive task, small child products do seem to cluster as a furniture type. The only two small child products completely outside the upper left quadrant are carry cots and baby nests; however, the dendrogram indicates these have similar enough overall fire risk and exposure profiles to cluster with play mats and bassinets.

In terms of CFR exposure, since small children are a group particularly susceptible to CFR exposure for several reasons, the potential for use of a type of furniture by a small child carries considerable weight in our model. This drives small child products together in the clustering, and up the Y axis in the scatter plot. In terms of fire risk, larger items used for resting in which people are more likely to smoke is a driver of clustering and pushes furniture types to the right of the X axis. Smoking was considered by the evaluators to be less of a risk for small child products as the children do not themselves smoke and are unlikely to be left unsupervised in a crib while smokers materials are around (the possibility of e.g. a cigarette being dropped into a crib

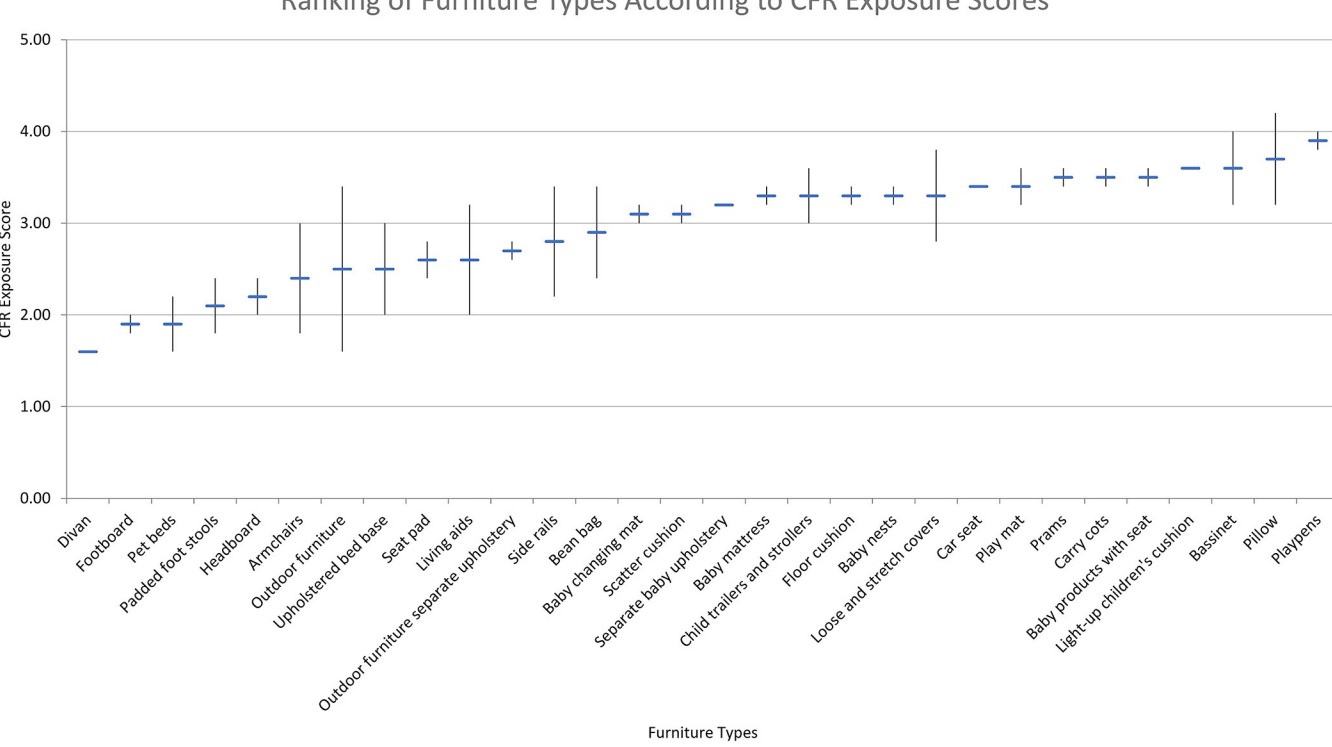

**Fig 11. Ranking of furniture types according to CFR exposure score.**

with a child, and remaining there while an adult is not present, cannot be eliminated, but was judged by the evaluators as relatively unlikely to occur compared to an adult smoking in an armchair and falling asleep). The two outlier small child products, baby nests and carry cots, are scored high for fire risk because they have a high number of junctions and are designed to be moved around, so were judged to be more likely to come into contact with smokers materials or candles.

In general, larger items of furniture present lower CFR exposure potential in our model. This should not be interpreted as meaning the absolute level of exposure to CFRs from large items of furniture is lower than for small child products. Large items will present a significant reservoir of CFRs that could be released to the environment over an extended period of time, and may pose particular issues during manufacture and disposal. They may also present particularly high risks if CFRs modify risk of mortality in a fire, e.g. through changes in production volume and density of smoke and toxic gases. However, because our model concerns furniture in use, issues such as behaviour of furniture during a fire or CFR release from disposal effectively have zero weight in our analysis.

## Strengths and limitations of the research

In an area in which empirical data is limited, working within restrictive time constraints, we believe we have nonetheless developed models and populated them with data that enables meaningful differentiation between types of domestic upholstered furniture based on anticipated fire risk and potential for CFR exposure. We achieved this by employing a mixed-methods approach, combining qualitative and quantitative methods to ground fire risk and CFR exposure models in the literature and expert opinion, in what we believe is a novel approach to

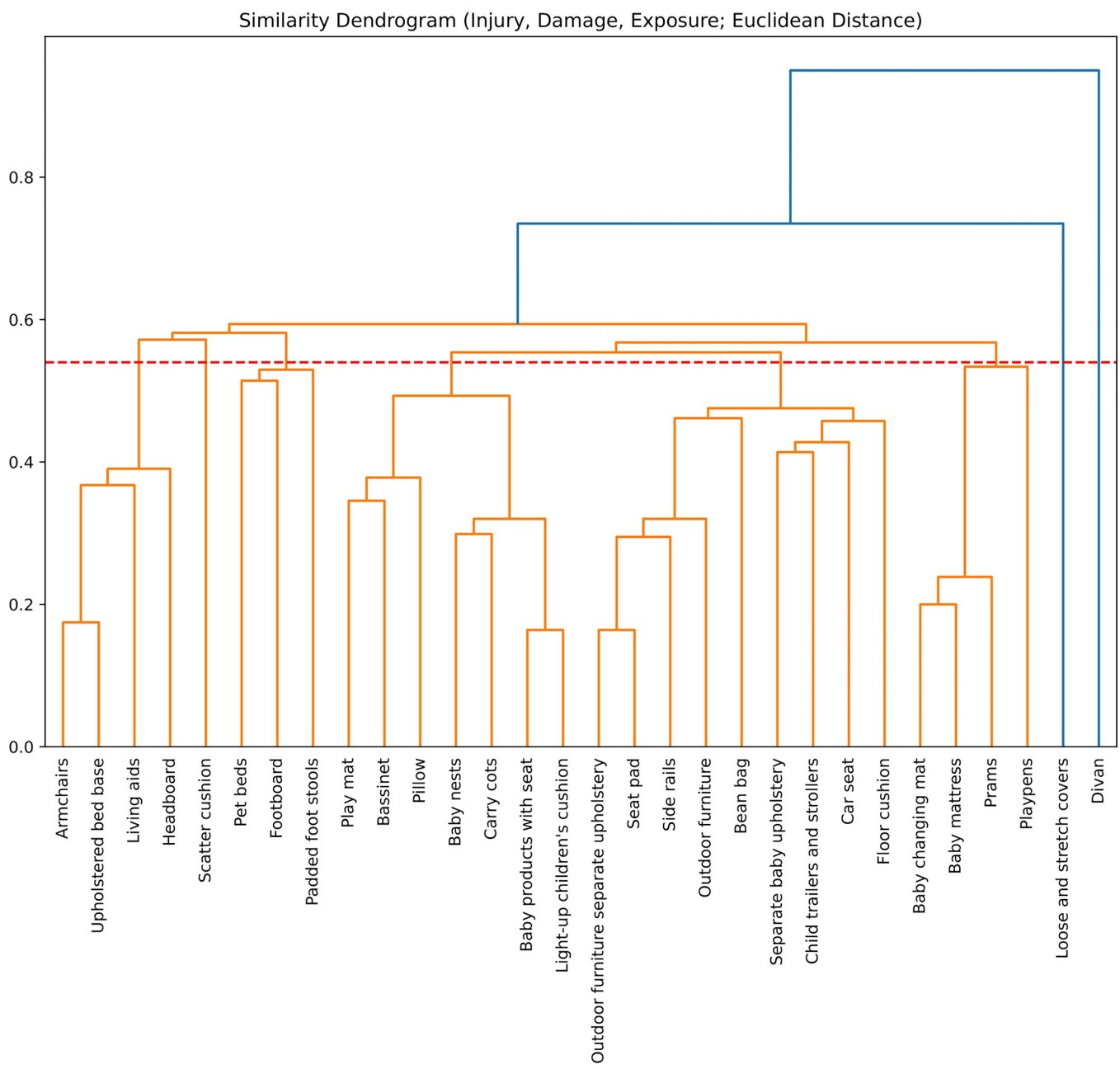

**Fig 12. Dendrogram of furniture types, with horizontal line drawn at 0.56.**

risk modelling in the fire safety sciences. Our methodology allowed us to integrate behavioural factors (likelihood of contact with an ignition source) and user vulnerabilities (ability to react to a fire in an item) when modelling risk of injury or damage in relation to fire in furniture. We believe these are novel additions to fire risk models.

**Availability of data and implications for improving furniture fire safety.** While a wide range of risk concepts are discussed in reviews of furniture fire safety literature, only a small proportion have been the subject of significant empirical investigation as to the role they play in fire risk. Fabric, foam, and upholstery are the most-mentioned terms in the primary

**Table 4. Furniture clusters as identified via dendrogram inspection. Small child products are in green, other products in blue.**

| Cluster 1 | Cluster 2 | Cluster 3 | Cluster 4 | Cluster 5 | Outliers |
|---|---|---|---|---|---|
| Playpens | Floor cushions | Light-up children's cushion | Padded foot stools | Headboards | Divans |
| Prams | Car seats | Baby products with seat | Footboards | Living aids | Loose and stretch covers |
| Baby mattresses | Child strollers and trailers | Carry cots | Pet beds | Upholstered bed bases | Scatter cushions |
| Baby changing mats | Baby separate upholstery | Baby nests | | Armchairs | |
| | Bean bags | Pillows | | | |
| | Outdoor furniture | Bassinets | | | |
| | Side rails | Play mats | | | |
| | Seat pads | | | | |
| | Outdoor furniture separate upholstery | | | | |

literature by some margin, and smokers materials are the most-studied source of ignition; however, the role of age and disability in risk of injury in fire is relatively unstudied (see S10 File, "Fire risk code thesaurus" and "Risk Term Frequency" sheets). In interview, fire investigators mentioned fuel poverty and drug or alcohol abuse as two major fire risk factors. The

**Fig 13. Two-dimensional plot of injury risk score and CFR exposure score for each furniture type, with quadrants.**

## Sensitivity of Fire Risk Scores to "Contact" Score Evaluations

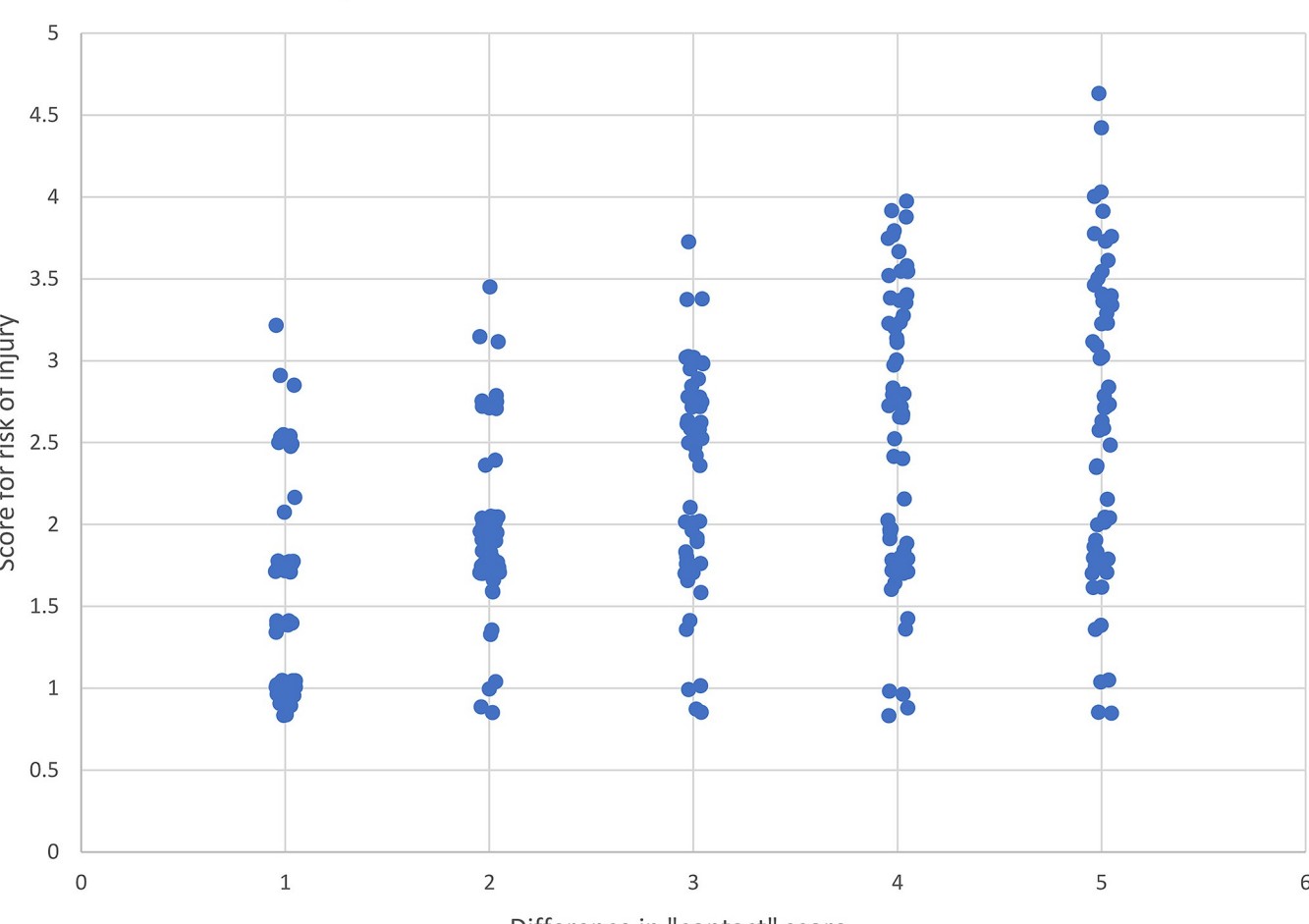

**Fig 14. Sensitivity of furniture type clustering to differences in contact score evaluations in the fire risk model.**

former results in fires due to accidents involving halogen heaters being used as a source of warmth that is cheap in comparison to central heating, the latter from erratic behaviour and vulnerability due to impaired situational awareness. This suggests there are specific vulnerable subpopulations that are beyond the level of resolution offered by our model (for example, it is possible that small child products are very low risk except in certain specific circumstances), and there are potentially important secondary sources of ignition (such as blankets igniting by being in very close proximity to halogen heaters) that are beyond the scope of our model.

**Scope and interpretation of the models.** The furniture fire safety matrix is designed to provide evidence to support policy decisions relating to the scope of the FFRs, based on anticipated CFR exposure and fire risk. Due to a lack of empirical data, it was necessary to model relative fire risk and relative potential for CFR exposure, based on data categories that could be expected to be reliably evaluated by informed persons. Low scores for exposure to CFRs from items of furniture should not, therefore, be interpreted as being of low potential concern. For example, headboards score relatively low, but may still pose an absolute level of CFR exposure that may be problematic. Furthermore, our model says nothing about what approaches ought to be taken to ensure that furniture is fire safe. Decisions about such approaches may need to

**Table 5. Comparison of advisory panel member evaluations with consolidated team evaluations ("Team, Consolidated") for each data input category in the fire risk model.**

| Ref No. | Furniture Type | Risk Variant | Evaluator | Contact with Ignition Source | Number of Junctions | Ornateness | Reactivity of Person |
|---|---|---|---|---|---|---|---|
| 30 | Armchair | High | Team, Consolidated | 5 | 3 | 3 | 5 |
| | | | Advisory Panel 1 | 4 | 3 | 1 | 5 |
| | | | Advisory Panel 2 | 4 | 3 | 2 | 4 |
| | | | Advisory Panel 3 | 4 | 5 | 4 | 4 |
| | | Low | Team, Consolidated | 3 | 1 | 2 | 2 |
| | | | Advisory Panel 1 | 1 | 1 | 1 | 1 |
| | | | Advisory Panel 2 | 2 | 1 | 2 | 1 |
| | | | Advisory Panel 3 | 3 | 1 | 3 | 1 |
| 1 | Headboard | High | Team, Consolidated | 4 | 3 | 3 | 4 |
| | | | Advisory Panel 1 | 3 | 1 | 3 | 3 |
| | | | Advisory Panel 2 | 2 | 0 | 2 | 2 |
| | | | Advisory Panel 3 | 4 | 0 | 5 | 1 |
| | | Low | Team, Consolidated | 3 | 1 | 1 | 4 |
| | | | Advisory Panel 1 | 1 | 1 | 1 | 3 |
| | | | Advisory Panel 2 | 2 | 0 | 1 | 2 |
| | | | Advisory Panel 3 | 4 | 0 | 3 | 1 |
| 14 | Pushchair | High | Team, Consolidated | 1 | 12 | 4 | 3 |
| | | | Advisory Panel 1 | 2 | 6 | 3 | 1 |
| | | | Advisory Panel 2 | 2 | 6 | 4 | 2 |
| | | | Advisory Panel 3 | 2 | 6 | 5 | |
| | | Low | Team, Consolidated | 1 | 3 | 3 | 3 |
| | | | Advisory Panel 1 | 2 | 3 | 3 | 1 |
| | | | Advisory Panel 2 | 2 | 1 | 3 | 2 |
| | | | Advisory Panel 3 | 3 | 3 | 2 | |
| 26 | Pet bed | High | Team, Consolidated | 4 | 3 | 2 | 3 |
| | | | Advisory Panel 1 | 1 | 1 | 1 | 1 |
| | | | Advisory Panel 2 | 1 | 1 | 1 | 1 |
| | | | Advisory Panel 3 | 2 | 1 | 4 | 2 |
| | | Low | Team, Consolidated | 3 | 0 | 1 | 2 |
| | | | Advisory Panel 1 | 1 | 0 | 1 | 1 |
| | | | Advisory Panel 2 | 0 | 0 | 0 | 1 |
| | | | Advisory Panel 3 | 3 | 4 | 3 | 2 |

account for the environmental or health implications of using potentially large quantities of CFRs to comply with fire safety tests, and the behaviour of furnishings in a fire, such as smoke opacity and toxic gas production. These are additional issues beyond the scope of our model.

## Recommendations for development of the fire risk and exposure models

Our main challenge was the time constraint we were working under to produce a functional model to support the revisions of the FFRs. This limited the amount of data we could collect and analyse, and the number of model assumptions that we could alter and test. We have provided baseline models for discussion, and a path forward for their development. We hope the level of detail we have provided in the methods and supplemental materials is sufficient to enable our models to be extended and/or modified.

**Domain identification and selection.** The concept networks show how lower-level fire risk and CFR exposure factors relate to higher-level factors, that the user can be confident is

grounded in the literature and expert opinion, and provides a basis for an informed discussion about the level of granularity desired from the model. To develop the networks, more literature could be reviewed; however, we would instead recommend prioritising extension and reorganisation of the concept network with more expert input. In particular, this should include analysis of narrative text from fire investigation reports and more interviews with non-academic domain experts such as fire investigators. These are two sources of concepts to which we did not have significant access and may provide additional perspectives not found in the published literature. This could impact choices about domain selection, domain weighting, and data input categories.

**Domain weighting and transformation functions.** The relative weight of each domain in the models could be finessed. For fire risk, differences in criticality of failure mode could be incorporated into the model. For potential exposure to CFRs, not all dimensions will contribute equally to exposure. There might also be interactions between domains. It may be the case, for example, that size of item, dermal migration, and duration of contact is cumulatively more important relative to oral migration when it comes to CFR exposure. A large item of furniture, as a large reservoir of FRs, may also present a disproportionately large source of exposure. We log transformed surface areas so smaller changes are more important when the surface area is small, in order to improve differentiation and increase clustering; however, a different transformation function may be more appropriate, especially if the objective is to accurately model relative potential for CFR exposure independent of the clustering objective we had in the present study. Finding or generating empirical data that will support weighting decisions will be challenging.

**Input data.** Model scores could be made more precise and more generalisable if more evaluations were conducted, and if empirical data could either be located or generated for model dimensions.

*Increasing the number of evaluations.* Increasing the number of evaluators involved in providing input data for data categories that involve subjective judgement would give a clearer indication of spread of subjective judgements. This would at least reduce uncertainty in the model due to potential variance in evaluator judgements, and it may increase precision of the model. Evaluating more types of furniture would present a more complete picture of the whole furniture landscape and could result in more robust clustering and/or extension of an identified cluster to other furniture types. Evaluating more items of furniture within a type would improve the precision of the model for each type. When increasing the number of evaluations, researchers should be aware that the evaluation process is time-intensive. Our training and consolidation process was designed to compensate for the small number of evaluators; increasing the number of evaluations without training may generate noisy data that does not improve the model.

*Adding empirical data.* Direct, empirical data for each domain could be generated through new primary studies or potentially derived from the literature, if available from a source that we did not identify or did not have capacity to analyse. Data on frequency of contact of furniture with ignition sources, and understanding the ignition source type (e.g. primary or secondary ignition source) could be especially useful for the model and informing risk management decisions, as would the role of the ability of people to react to a fire in relation to an item of furniture in understanding risk. Tracing CFR exposure back to specific types of furniture would improve the exposure model but may be difficult to achieve.

## Conclusions

Despite working with very limited empirical data, we were able to develop models for furniture fire risk and CFR exposure potential that differentiated between furniture types. We believe we

offer the first model reconciling fire risk with CFR exposure, and one of the first that takes into account user behaviour in modelling fire risk. We identified a meaningful cluster of furniture types sharing relatively high potential for CFR exposure with relatively low risk of causing injury in a fire initiated in the item. This cluster included a number of baby and infant products plus pillows.

Given the scope of the research, that the models were developed to support identification of clusters of furniture types, and that the models provide relative rather than absolute measures of fire risk and CFR exposure, it follows that users should be careful not to over-interpret the furniture rankings we present. Nonetheless, the models should be valuable for informing future fire risk and CFR exposure modelling in relation to furniture. In particular, our work has highlighted a preponderance of empirical data on fabric and foam flammability in response to contact with small open flames and smokers materials, and a lack of research being conducted in a range of factors that also influence fire risk in furniture. In order to be more fully informative of interventions to improve furniture fire safety, fire science needs to move beyond testing material combinations and toward conducting empirical research into other fire risk factors.

## Supporting information

**S1 File. Index of supplemental materials.**
(PDF)

**S2 File. List of documents included in the literature review, with the number of concepts identified in each document.**
(XLSX)

**S3 File. The annotation guidelines manual, describing the document annotation methods and processes.**
(PDF)

**S4 File. High-level summary concept network, for concepts relating to fire risk.**
(PDF)

**S5 File. Concept network integrating concepts of fire risk and upholstered furniture.**
(PDF)

**S6 File. High-level summary concept network, for exposure to CFRs.**
(PDF)

**S7 File. Complete concept network for exposure to CFRs.**
(PDF)

**S8 File. Concept network integrating concepts of CFR exposure and upholstered furniture.**
(PDF)

**S9 File. Concept network integrating concepts of fire risk, CFR exposure, and upholstered furniture.**
(PDF)

**S10 File. Code list for fire risk and CFR exposure models.**
(XLSX)

**S11 File. Literature analysis for 3385 citations as an estimate of conceptual coverage of existing empirical evidence relating to modelling objectives.**
(XLSX)

**S12 File. Evaluation questionnaire for assessing fire risk posed by specific items of small upholstered furniture (blank).**
(XLSX)

**S13 File. Evaluation questionnaire for assessing fire risk posed by specific items of small upholstered furniture, with volume calculations.**
(XLSX)

**S14 File. Evaluation questionnaire for assessing potential for CFR exposure from specific items of small upholstered furniture (blank).**
(XLSX)

**S15 File. Evaluation questionnaire for assessing potential for CFR exposure from specific items of small upholstered furniture, with surface area calculations.**
(XLSX)

**S16 File. Raw data, analysis, and visualisations for the fire risk, CFR exposure, and integrated risk-exposure models.**
(XLSX)

**S17 File. Explanation of shortage of empirical data for CFR exposure model.**
(DOCX)

**S18 File. Hi-resolution dendrogram for fire risk clusters.**
(PDF)

**S19 File. High-resolution scatter plot of fire risk against CFR exposure, with quadrants.**
(PDF)

**S20 File. Sensitivity analysis for model results.**
(XLSX)

**S21 File. Risk score calculations pseudocode.**
(DOCX)

## Acknowledgments

The authors would like to thank Grunde Jomaas, Baljinder Kandola, Phil Reynolds, Clive Steel, and the London Fire Brigade for constructive comments on the methodology presented.

## Author Contributions

**Conceptualization:** Paul Whaley, Nia Bell, Stuart Harrad, Thomas Kirkbride, Dzhordzhio Naldzhiev, T. Richard Hull.

**Data curation:** Paul Whaley, Stephen Wattam.

**Formal analysis:** Paul Whaley, Stephen Wattam.

**Investigation:** Paul Whaley, Clare Bedford, Nia Bell, Stuart Harrad, Nicola Jones, Elena Payne, Elli-Jo Wooding, T. Richard Hull.

**Methodology:** Paul Whaley, Stephen Wattam.

**Project administration:** Paul Whaley, T. Richard Hull.

**Software:** Stephen Wattam.

**Supervision:** Thomas Kirkbride, Dzhordzhio Naldzhiev, T. Richard Hull.

**Visualization:** Paul Whaley, Stephen Wattam.

**Writing – original draft:** Paul Whaley, Nia Bell, Stuart Harrad, T. Richard Hull.

**Writing – review & editing:** Paul Whaley, Stephen Wattam, Nia Bell, Stuart Harrad, Thomas Kirkbride, Dzhordzhio Naldzhiev, T. Richard Hull.

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
