## [Decision Letter · Decision Letter 0]

4 Sep 2023

PONE-D-23-24715Reconciling chemical flame retardant exposure and fire risk in domestic furniturePLOS ONE

Dear Dr. Whaley,

Thank you for submitting your manuscript to PLOS ONE. After careful consideration, we feel that it has merit but does not fully meet PLOS ONE’s publication criteria as it currently stands. Therefore, we invite you to submit a revised version of the manuscript that addresses the points raised during the review process.

We look forward to receiving your revised manuscript.

Kind regards,

Yanping Yuan

Academic Editor

PLOS ONE

Journal Requirements:

When submitting your revision, we need you to address these additional requirements. 1. Please ensure that your manuscript meets PLOS ONE's style requirements, including those for file naming. The PLOS ONE style templates can be found at https://journals.plos.org/plosone/s/file?id=wjVg/PLOSOne_formatting_sample_main_body.pdf and " ext-link-type="uri" xlink:type="simple">https://journals.plos.org/plosone/s/file?id=ba62/PLOSOne_formatting_sample_title_authors_affiliations.pdf" 2. Thank you for stating the following financial disclosure:  [The research was funded under competitive tender "CR21021 - Fire Risks of Upholstered Products" as part of the UK Department for Department for Business, Energy Industrial Strategy (BEIS) Office for Product Safety and Standards (OPSS) Strategic Research Programme. The contract was awarded to University of Central Lancashire with TRH as Principal Investigator. PW, SW, NB, SH, and EP were engaged as subcontractors. TK and DN were involved as BEIS employees in study design and preparation of the manuscript. The manuscript went through an internal clearance process at BEIS and OPSS before being approved for submission.]Please state what role the funders took in the study.  If the funders had no role, please state: ""The funders had no role in study design, data collection and analysis, decision to publish, or preparation of the manuscript."" If this statement is not correct you must amend it as needed. Please include this amended Role of Funder statement in your cover letter; we will change the online submission form on your behalf. 3. We note that Figures 3 and 4 in your submission contain copyrighted images. All PLOS content is published under the Creative Commons Attribution License (CC BY 4.0), which means that the manuscript, images, and Supporting Information files will be freely available online, and any third party is permitted to access, download, copy, distribute, and use these materials in any way, even commercially, with proper attribution. For more information, see our copyright guidelines: http://journals.plos.org/plosone/s/licenses-and-copyright. We require you to either (1) present written permission from the copyright holder to publish these figures specifically under the CC BY 4.0 license, or (2) remove the figures from your submission: 1.) You may seek permission from the original copyright holder of Figures 3 and 4 to publish the content specifically under the CC BY 4.0 license.  We recommend that you contact the original copyright holder with the Content Permission Form (http://journals.plos.org/plosone/s/file?id=7c09/content-permission-form.pdf) and the following text:“I request permission for the open-access journal PLOS ONE to publish XXX under the Creative Commons Attribution License (CCAL) CC BY 4.0 (http://creativecommons.org/licenses/by/4.0/). Please be aware that this license allows unrestricted use and distribution, even commercially, by third parties. Please reply and provide explicit written permission to publish XXX under a CC BY license and complete the attached form.” Please upload the completed Content Permission Form or other proof of granted permissions as an ""Other"" file with your submission.  In the figure caption of the copyrighted figure, please include the following text: “Reprinted from [ref] under a CC BY license, with permission from [name of publisher], original copyright [original copyright year].” 2.) If you are unable to obtain permission from the original copyright holder to publish these figures under the CC BY 4.0 license or if the copyright holder’s requirements are incompatible with the CC BY 4.0 license, please either i) remove the figure or ii) supply a replacement figure that complies with the CC BY 4.0 license. Please check copyright information on all replacement figures and update the figure caption with source information. If applicable, please specify in the figure caption text when a figure is similar but not identical to the original image and is therefore for illustrative purposes only. 4. Please upload a copy of Figure 15, to which you refer in your text on page 32. If the figure is no longer to be included as part of the submission please remove all reference to it within the text. 5.Thank you for stating the following in the Competing Interests/Financial Disclosure section: "PW declares they are a self-employed consultant who provides research, training, and editorial services on a contracted basis for industry, NGO, academic, and publishing clients. Given the controversial nature of the issues researched in this project, the way the work is received could affect their prospects for securing future contracts in this area. However, they do not identify any specific financial interests that would be directly affected at this time. For non-financial interests, they declare involvement in environmental advocacy for over 15 years and being motivated by a perceived need for significant improvement in identification and management of potential risks to health posed by the environment in which humans live. They have written several papers and held public positions, such as developing and/or signing consensus statements, about the need for improved regulation and scientific assessment of chemical substances with flame retardants as an example. They are actively involved with NGOs, particularly in the UK, who are advocating for reforms to chemical regulation. Being prominently involved as a Research Fellow in the Evidence-Based Toxicology Collaboration, their views could be perceived as upholding that organisation's views. They are also Editor-in-Chief of Evidence-Based Toxicology, a new journal concerned with promoting improved research and publishing practices in toxicology and environmental health. While these financial and non-financial interests are pertinent to the manuscript, PW does not believe them to have compromised the integrity of the work undertaken. SW declares scientific consultancy services in fields unrelated to the topic of research in this manuscript. The way the work is received could affect their prospects of future employment. CB declares employment at the University of Central Lancashire in the Centre for Fire and Hazards Sciences as part of a team who examine the toxicity of flammable materials. CB is also a trade union representative at the University. As a result of previous work in their current role, CB declares an awareness of the potential hazards presented by both flammable materials and flame retardants, and the belief that it is important to accurately assess hazards and strive to protect people from them as much as possible. They do not think that this conviction has affected their ability to contribute objectively to this work. SH declares being an academic researcher. Given the controversial nature of the issues we have researched in this project, the way the work is received could affect their prospects for securing future research grants and contracts, as well external consultancy work in this area. However, they cannot identify any specific financial interests that would be directly affected at this time. For non-financial interests, their extensive previous research into human exposure to a wide range of chemical contaminants including flame retardants inherently influences the submitted publication. They declare being a current member of the UK government’s Hazardous Substances Advisory Committee; while this manuscript may be perceived as reflecting the views of this committee, it has not to date involved consideration of human exposure to flame retardants and moreover SH's views are purely their own and not those of the committee. While these financial and non-financial interests are pertinent to the manuscript, SH does not believe them to have compromised the integrity of the work undertaken. NJ declares employment as a Post Doctoral Researcher at the University of Central Lancashire at the time the research was carried out. TK declares they are currently employed by the Department for Business, Energy, and Industrial Strategy, and that they commissioned this work on behalf of the Office for Product Safety and Standards. They were involved in setting the objectives and the design of the research. As a civil servant they state they are impartial to the findings of the research, and that the research does not necessarily represent the views of the department. DN declares being employed by OPSS during the conduct of the study, as Head of the Science Strategy team overseeing research projects. EW declares employment by the University of Central Lancashire at the time the research was carried out. RH declares employment as a Professor of Chemistry and Fire Science by the University of Central Lancashire since 2007. They do not have any secondary employment, consultancy, board membership, patents or patent applications. They have received research funding from external organisations: in 2018, £15,000 by Silentnight Beds Ltd., to undertake four large scale fire tests and assess the smoke toxicity; in 2019, £161,000 from Innovate UK and Silentnight Beds Ltd., to support a Knowledge Transfer Partnership aimed at helping Silentnight reduce their fire retardant use, the smoke toxicity, and the recyclability of their products, in order to improve their access to the wider mainland European market (Silentnight Beds had no involvement in the current BEIS project, nor in the authorship of this paper); funding from Fire Safe Europe to support a PhD student investigating smoke toxicity of construction products; and funding from the Construction Products Group, Europe to develop fire protective coatings for structural steel. RH has also supported their colleague Prof Anna Stec in their role as an expert witness to the United Kingdom's Grenfell Tower Inquiry, and is a member of the BEIS Expert Advisory Panel for the revision of the English Furniture Flammability Regulations. RH does not believe they have any interests that compete with those of the research described in the manuscript."   We note that one or more of the authors are employed by a commercial company: University of Central Lancashire, Department for Business, Energy, and Industrial Strategy, OPSS,   1.) Please provide an amended Funding Statement declaring this commercial affiliation, as well as a statement regarding the Role of Funders in your study. If the funding organization did not play a role in the study design, data collection and analysis, decision to publish, or preparation of the manuscript and only provided financial support in the form of authors' salaries and/or research materials, please review your statements relating to the author contributions, and ensure you have specifically and accurately indicated the role(s) that these authors had in your study. You can update author roles in the Author Contributions section of the online submission form. Please also include the following statement within your amended Funding Statement. “The funder provided support in the form of salaries for authors [insert relevant initials], but did not have any additional role in the study design, data collection and analysis, decision to publish, or preparation of the manuscript. The specific roles of these authors are articulated in the ‘author contributions’ section.”If your commercial affiliation did play a role in your study, please state and explain this role within your updated Funding Statement.  2.) Please also provide an updated Competing Interests Statement declaring this commercial affiliation along with any other relevant declarations relating to employment, consultancy, patents, products in development, or marketed products, etc.   Within your Competing Interests Statement, please confirm that this commercial affiliation does not alter your adherence to all PLOS ONE policies on sharing data and materials by including the following statement: ""This does not alter our adherence to  PLOS ONE policies on sharing data and materials.” (as detailed online in our guide for authors http://journals.plos.org/plosone/s/competing-interests) . If this adherence statement is not accurate and  there are restrictions on sharing of data and/or materials, please state these. Please note that we cannot proceed with consideration of your article until this information has been declared. Please include both an updated Funding Statement and Competing Interests Statement in your cover letter. We will change the online submission form on your behalf. 6. We note that you have stated that you will provide repository information for your data at acceptance. Should your manuscript be accepted for publication, we will hold it until you provide the relevant accession numbers or DOIs necessary to access your data. If you wish to make changes to your Data Availability statement, please describe these changes in your cover letter and we will update your Data Availability statement to reflect the information you provide. 7. We note you have included a table to which you do not refer in the text of your manuscript. Please ensure that you refer to Tables 3 and 6 in your text; if accepted, production will need this reference to link the reader to the Table. 8. We notice that your supplementary "INDEX OF SUPPLEMENTAL MATERIALS" are included in the manuscript file. Please remove them and upload them with the file type 'Supporting Information'. Please ensure that each Supporting Information file has a legend listed in the manuscript after the references list.

Reviewers' comments:

Reviewer's Responses to Questions

**Comments to the Author**

1. Is the manuscript technically sound, and do the data support the conclusions?

Reviewer #1: Yes

Reviewer #2: Yes

2. Has the statistical analysis been performed appropriately and rigorously? 

Reviewer #1: Yes

Reviewer #2: Yes

3. Have the authors made all data underlying the findings in their manuscript fully available?

Reviewer #1: Yes

Reviewer #2: Yes

4. Is the manuscript presented in an intelligible fashion and written in standard English?

Reviewer #1: Yes

Reviewer #2: Yes

5. Review Comments to the Author

Reviewer #1: Overall evaluation

The topic of furniture as exposure source of hazardous chemicals is of contemporary interest for the research community, for policy makers and the related industries as well as for the public.

The topic is also in the scope of the journal.

The paper can in my opinion be accepted after minor revisions.

Major comments

You lump all chemical flame retardants together which is an oversimplification. Please mention that there are e.g. halogenated flame retardants which are considered particular problematic. Possibly mention the review article of Shaw et al. 2010.

Shaw SD, Blum A, Weber R, Kannan K, Rich D, Lucas D, Koshland CP, Dobraca D, Hanson S, Birnbaum LS. (2010) Halogenated Flame Retardants: Do the Fire Safety Benefits Justify the Risks? Rev. Environ. Health 25(4), 261-305.

Please mention that almost all other countries do not have this flame retardant standard which require that the bulk polymer in furniture is flame retarded and that in particular UK and US had this stringent standard. Please mention that US/California modified their flammability standard for furniture which does not require to use flame retardants for the bulk polymer/PUR now. Please reference to Charbonnet et al. 2020 and possibly others.

Charbonnet, J.A., Weber, R. and Blum, A., 2020. Flammability standards for furniture, building insulation and electronics: Benefit and risk. Emerging Contaminants, 6, pp.432-441.

Please mentionthe situation of other European countries (also almost all other countries): Most of them do not use flame retardants in PUR foam in furniture. I do not think that they experience unacceptable fire risks. Please mention and discuss. If possible please compare fire death or fire frequency in at least some European countries which do not require flame retardants in bulk materials in furniture with UK.

Specific comments on sections and paragraphs:

Introduction

Line 100: “The most widely used CFRs in domestic furniture include decabromodiphenyl ethane (DBDPE) “. Please mention that this is today and that in the past decaBDE was mainly used. Also please mention that PBDEs are listed as POPs in the Stockholm Convention for global phase out.

Line 107: “DBDPE to Deca-BDE suggests that it too will be withdrawn at some point in the near future: it is currently under assessment as PBT (ECHA, 2018).” Please mention here the new proposed strategy of ECHA for flame retardants with suggestion to phase out aromatic brominated flame retardants as a group.

Line 117ff “Meanwhile, pressures such as reducing the presence of toxic substances in our environment, the need to recycle products such as furniture at their end-of-life, and the “unsuitability of furniture containing CFRs for alternative end-of-life processes, such as landfill or material reclamation, are much higher priorities than they were in the 1980s.” Not clear – please rephrase.

Result and discussion

Line 387ff: “We do, however, know that none of the reviews in our document set that addressed human exposure to flame retardants discussed in detail how specific items of

furniture contribute to a person’s exposure to CFRs.”

Abdallah Harrad (2018) have conducted studies which showed that HBCD in furniture textiles result in relevant exposure by skin contact. Please mention this and reference to

Abdallah, M.A.E. and Harrad, S., 2018. Dermal contact with furniture fabrics is a significant pathway of human exposure to brominated flame retardants. Environment international, 118, pp.26-33.

Also Pawar, G., Abdallah, M.A.E., de Sáa, E.V. and Harrad, S., 2017. Dermal bioaccessibility of flame retardants from indoor dust and the influence of topically applied cosmetics. Journal of exposure science environmental epidemiology, 27(1), pp.100-105.

You mention the first paper in the Table 3 but suggest to mention it also in the text.

Table 3: (Abou-Elwafa Abdallah and Harrad, 2022). Think this is (Abdallah and Harrad, 2022). Also in his researchgate he abbreviate to Mohammed A. Abdallah)

Table 5: Baby mattress has lower score (3.4) compare to Baby products with seat (3.6) and Bassinet (4.0). However mattress has likely the by far longest contact time. Possibly modify?

Line 566: “moved around, so were judged to be more likely to come into contact with smokers materials.” While smokers materials are certainly relevant, suggest also to mention candles having high use frequency.

Reviewer #2: The manuscript is exceptionally well-written, providing valuable insights into future fire risk assessment and CFR exposure modeling concerning furniture. The comprehensive explanation and presentation of limited information and empirical data regarding the flammability of fabric and foam when exposed to small open flames and materials used by smokers are noteworthy. There are no discernible flaws in this publication, and it is ready for publication as is.

6. PLOS authors have the option to publish the peer review history of their article (what does this mean?). If published, this will include your full peer review and any attached files.

Reviewer #1: No

Reviewer #2: No

---

## [Author Response · Author response to Decision Letter 0]

21 Sep 2023

See "response to reviewers" document as uploaded.

---

## [Decision Letter · Decision Letter 1]

18 Oct 2023

Reconciling chemical flame retardant exposure and fire risk in domestic furniture

PONE-D-23-24715R1

Dear Dr. Whaley,

We’re pleased to inform you that your manuscript has been judged scientifically suitable for publication and will be formally accepted for publication once it meets all outstanding technical requirements.

Kind regards,

Yanping Yuan

Academic Editor

PLOS ONE

Additional Editor Comments (optional):

Reviewers' comments:

Reviewer's Responses to Questions

**Comments to the Author**

1. If the authors have adequately addressed your comments raised in a previous round of review and you feel that this manuscript is now acceptable for publication, you may indicate that here to bypass the “Comments to the Author” section, enter your conflict of interest statement in the “Confidential to Editor” section, and submit your "Accept" recommendation.

Reviewer #1: All comments have been addressed

2. Is the manuscript technically sound, and do the data support the conclusions?

Reviewer #1: Yes

3. Has the statistical analysis been performed appropriately and rigorously? 

Reviewer #1: N/A

4. Have the authors made all data underlying the findings in their manuscript fully available?

Reviewer #1: Yes

5. Is the manuscript presented in an intelligible fashion and written in standard English?

Reviewer #1: Yes

6. Review Comments to the Author

Reviewer #1: The topic of furniture as exposure source of hazardous chemicals is of contemporary interest for the research community, for policy makers and the related industries as well as for the public.

The topic is also in the scope of the journal.

The paper can be accepted after reviewer comments have been addressed or replies have been given.

7. PLOS authors have the option to publish the peer review history of their article (what does this mean?). If published, this will include your full peer review and any attached files.

Reviewer #1: No

---

## [Editor Report · Acceptance letter]

25 Oct 2023

PONE-D-23-24715R1 

Reconciling chemical flame retardant exposure and fire risk in domestic furniture 

Dear Dr. Whaley:

I'm pleased to inform you that your manuscript has been deemed suitable for publication in PLOS ONE. Congratulations! Your manuscript is now with our production department. 

Kind regards, 

on behalf of

Prof. Yanping Yuan 

Academic Editor

PLOS ONE